# Needles in the Haystack: Addressing Signal Dilution Improves scRNA-seq Perturbation Response Modeling and Evaluation

Gabriel M. Mejia [* 1]   Henry E. Miller [* 1]   Francis J. A. Leblanc [1]   Bo Wang [2]   Brendan Swain [1]
Lucas Paulo de Lima Camillo [1]

## Abstract

Recent benchmarks reveal that single-cell perturbation response models are often outperformed by simply predicting the dataset mean. Through large-scale *in silico* simulations, together with analyses of two real-world perturbation datasets, we trace this anomaly to a metric artifact: unweighted error metrics systematically reward mean predictions when perturbation effects are sparse. To address this limitation, we introduce differentially expressed gene (DEG)-aware metrics—weighted mean-squared error (WMSE) and weighted delta $R^2$ ($R_w^2(\Delta)$)—that sensitively measure error in niche, perturbation-specific signals. We further propose explicit negative and positive performance baselines to calibrate these metrics. Under this framework, the mean baseline sinks to null performance, while genuinely informative predictors are correctly rewarded. Finally, we show that using WMSE as a training objective reduces mode collapse and improves predictive performance across multiple model architectures.

## 1. Introduction

In recent years, advances in CRISPR-based screening and single-cell RNA sequencing (scRNA-seq) have enabled large-scale perturbation datasets (Peidli et al., 2024) that capture transcriptomic responses of individual cells to genetic perturbations. Models trained on such data promise *in silico* perturbation screening, potentially accelerating therapeutic discovery by identifying interventions that restore function to cells in disease states. In response, a wide range of perturbation-response models have been proposed, span-

*Equal contribution   [1]Shift Bioscience, Cambridge, UK   [2]University of Toronto, Vector Institute, Toronto, Canada. Correspondence to: Lucas Paulo de Lima Camillo <lucas@shiftbioscience.com>.

*Proceedings of the $43^{rd}$ International Conference on Machine Learning*, Seoul, South Korea. PMLR 306, 2026. Copyright 2026 by the author(s).

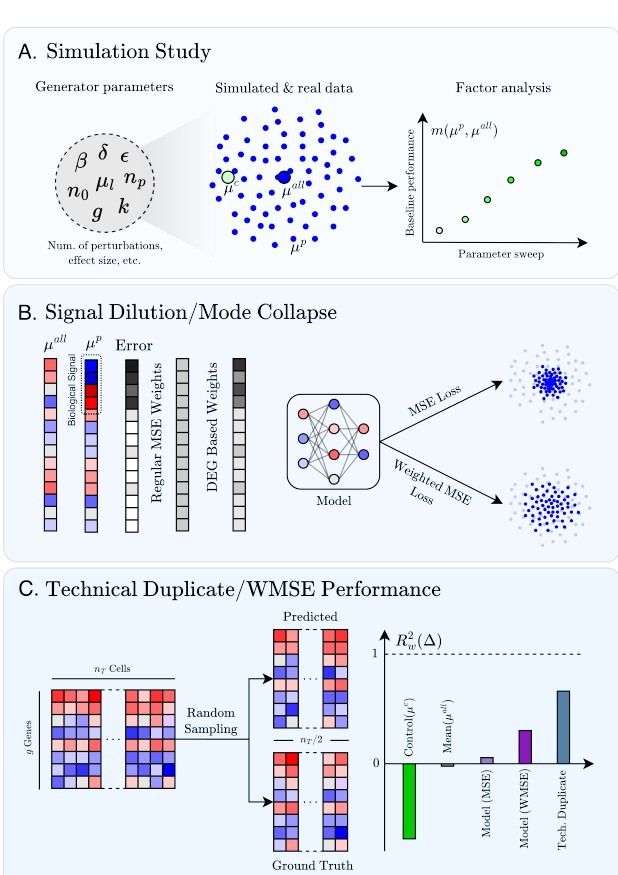

*Figure 1.* (A) Using a fixed set of generator parameters we compute both simulated and real datasets with specific characteristics. We then sweep over parameters to uncover the factors influencing baseline performance on different metrics. (B) True biological signal is diluted in scRNA-seq data causing mode collapse in model predictions. Introducing biologically aware weights (WMSE) ameliorates this problem. (C) We posit the technical duplicate as the optimal positive control for benchmarking, and show that WMSE as a training objective improves model performance compared to MSE.

ning optimal transport (CellOracle (Kamimoto et al., 2023)), prior knowledge graph learning (GEARS (Roohani et al., 2024)), and transformer-based foundation models (scGPT (Cui et al., 2024), STATE (Adduri et al., 2025)).

Despite this rapid methodological progress, recent benchmarking studies have reported a troubling pattern: simple

predictors, such as linear models, frequently outperform more sophisticated architectures (Li et al., 2024a; Wu et al., 2026; Csendes et al., 2025; Li et al., 2024b; Bendidi et al., 2024; Wenteler et al., 2025; Ahlmann-Eltze et al., 2025). More strikingly, an untrained mean baseline—defined as predicting the average expression profile of all perturbed cells in the training set, irrespective of perturbation identity—not only achieves strong performance under standard evaluation metrics, but often surpasses most deep learning models without learning any perturbation-specific information.

Motivated by these findings, we ask a simple question: *Why does the mean baseline perform so well under standard metrics?* Using a large-scale simulation study (Fig. 1a), we identify several dataset properties that inflate mean-baseline performance, with *signal dilution*—true biological changes confined to a small fraction of genes (Fig. 1b)—emerging as a dominant driver. To address this, we introduce evaluation metrics sensitive to niche signals and a technical duplicate baseline that provides a realistic positive performance reference (Fig. 1c). Finally, we show that accounting for signal dilution during training via a weighted loss reduces mode collapse and improves predictive performance across models and datasets. Our key contributions are summarized as follows:

1. We identify signal dilution as a key contributor to the overestimated performance of the mean baseline.

2. We introduce metrics and baselines to account for the signal dilution problem in model evaluations.

3. We propose a simple, drop-in replacement loss that decreases mode collapse and improves performance across models and datasets.

Together, these advances offer avenues for iterative improvement and allow more transparent assessment of perturbation response models. The code for this work is found at https://github.com/shiftbioscience/Needles-In-The-Haystack.

## 2. Background

### 2.1. Single-cell Perturbation Data

Single-cell RNA sequencing (scRNA-seq) measures transcript abundances in individual cells, producing a sparse (typically $> 90\%$ zeros) gene-by-cell count matrix $\mathbf{X} \in \mathbb{N}^{g \times n_T}$ that is well modeled by a negative-binomial distribution. Standard preprocessing includes library-size normalization, log transformation, and selection of $\sim$ 2–5k most highly variable genes.

Large-scale genetic perturbation screens combine CRISPR

interventions with scRNA-seq, yielding paired control–perturbation observations suitable for supervised learning. The Perturb-seq screening technology knocks out, represses (CRISPRi) or activates (CRISPRa) target genes prior to measurement of gene expression (Dixit et al., 2016). Perturb-seq datasets are often used to train perturbation response models that aim to generalize to unseen perturbations or tissue types, with the hope of enabling large-scale *in silico* screens where millions of perturbations can be tested without the need for costly and time-consuming wet lab experiments.

### 2.2. Perturbation Response Models

Predictive models fall into four archetypes. (i) Simple linear baselines: ridge or principal-component regression that extrapolate from control and single-perturbation means. (ii) Autoencoder-based models: scGen, CPA and scVI fine-tune autoencoders to encode a cell and an intervention; counterfactuals are obtained by vector arithmetic in latent space (Lotfollahi et al., 2019; 2023; Lopez et al., 2018); (iii) Prior knowledge graph learning: GEARS learns gene embeddings on a co-expression graph and perturbation embeddings on a gene-ontology graph, then decodes their interaction to predict expression shifts (Roohani et al., 2024). (iv) Transformer-based foundation models: scGPT (Cui et al., 2024) and STATE (Adduri et al., 2025) pre-train on millions of cells to learn fundamental gene-gene relationships that inform perturbation response prediction. Although every paradigm reflects key assumptions and inductive biases, they all strive to learn a conditional generation function $\hat{\mathbf{X}}^p = f(p, \mathbf{X^c})$ that predicts how the control cell population $\mathbf{X^c}$ would respond to perturbation $p$.

### 2.3. Common Metrics and Mean Predictors

Performance is typically evaluated at the *pseudobulk* level by averaging predicted and ground-truth single-cell profiles per perturbation. This aggregation mitigates scRNA-seq sparsity and reduces the task to average effect prediction. Among many proposed metrics, the most commonly reported are MSE (or MAE) and Pearson($\Delta$). MSE measures the average $L_2$ error, while Pearson($\Delta$) assesses the correlation between predicted and observed changes relative to a control mean profile $\mu^c$. For a perturbation with mean profile $\mu^p$ and prediction $\hat{\mu}^p$, Pearson($\Delta$) is defined as $r(\mu^p - \mu^c, \hat{\mu}^p - \mu^c)$.

Multiple recent benchmarks that employ these and other metrics have independently found that predicting the mean baseline (the mean of all perturbed cells; $\mu^{all}$) often matches or surpasses state-of-the-art models. Li et al. (2024a) assessed ten methods across multiple modeling tasks where the mean baseline achieved the lowest MAE($\Delta$) and nearly the highest Pearson($\Delta$). Analyzing four Perturb-

*Table 1.* Parameter space used for simulation and real data experiments.

| Parameter | Effect | Range Simulation | Range *Norman19* | Range *Replogle22* |
|---|---|---|---|---|
| $g$ | Number of genes in the dataset | Linear: $1000 - 8192$ | $\log_2$: $2 - 8192$ | $\log_2$: $2 - 8192$ |
| $n_0$ | Number of control cells | log: $10 - 8192$ | $\log_2$: $1 - 8192$ | $\log_2$: $1 - 8192$ |
| $n_p$ | Number of cells per perturbation | log: $10 - 256$ | $\log_2$: $2 - 256$ | $\log_2$: $2 - 64$ |
| $k$ | Number of perturbations per dataset | log: $10 - 2000$ | $\log_2$: $1 - 175$ | $\log_2$: $1 - 1334$ |
| $\beta$ | Amount of systemic bias in the control | Linear: $0 - 2$ | Linear: $0 - 2$ | Linear: $0 - 2$ |
| $\delta$ | Probability of perturbing a gene | Linear: $0.001 - 0.1$ | Quantiles: $0 - 1$ | Quantiles: $0 - 1$ |
| $\epsilon$ | Multiplicative effect of perturbations | log: $1.2 - 5.0$ | $-$ | $-$ |
| $\mu_l$ | Library size scaling (Data quality) | log: $0.2 - 5.0$ | Deciles: $0 - 1$ | Deciles: $0 - 1$ |

seq datasets, Csendes et al. (2025) found that the mean baseline exceeds scGPT and scFoundation on Pearson($\Delta$). Wenteler et al. (2025) showed that the mean baseline tracks top-20 DEG effects as closely as scGPT, Geneformer, or UCE (Cui et al., 2024; Theodoris et al., 2023; Rosen et al., 2023). Finally, Ahlmann-Eltze et al. (2025) reported that four foundation and two deep-learning models fail to beat the mean baseline in evaluations across multiple datasets.

Recent work has shown that Pearson($\Delta$) scores for the mean baseline can be artificially inflated by control-cell bias (Viñas Torné et al., 2025)—the degree to which control cells differ from the average perturbation effect. However, less is known about which additional dataset and metric properties contribute to its strong performance under this metric. Moreover, the factors that cause the mean baseline to perform competitively under common error-based metrics such as MSE and MAE are not yet well characterized.

## 3. Methods

### 3.1. *In silico* Simulations

We model synthetic datasets containing $n_0$ control cells and $k$ perturbations with a constant number of $n_p$ observed cells per perturbation. Each cell is represented by a random raw count vector $X \in \mathbb{R}^g$ where each component represents the observed expression of a single gene in that cell under a unique perturbation. Mathematically, we model the raw count expression value $X_{i,j}^p$ of the $i^{th}$ gene of the $j^{th}$ cell belonging to perturbation $p$ as a negative binomial with a fixed per-gene dispersion and variable mean:

$$X_{i,j}^p \sim \text{NB}\left(\mu_{i,j}^p, \theta_i\right) \quad (1)$$

Where $\theta_i$ is the fixed dispersion of gene $i$ and $\mu_{i,j}^p$ captures the simulated perturbation effects as follows:

$$\text{Perturbations:} \quad \mu_{i,j}^p = l_j \alpha_i^p (\mu_i^c + \beta \lambda_i) \quad (2)$$

$$\text{Control:} \quad \mu_{i,j}^c = l_j \mu_i^c \quad (3)$$

$\mu_i^c$ being the average of control expression, $\lambda_i$ a scalar symbolizing a realistic systematic bias between the control pop-

ulation and all other perturbations (only depends on the gene), $\beta$ a global dataset parameter controlling the severity to which $\lambda_i$ is applied (zero for a perfectly centered control), $\alpha_i^p$ a multiplicative effect on gene $i$ associated with perturbation $p$, and $l_j$ the library size component and affects every gene of the cell $j$ equally. Both the perturbation effect $\alpha_i^p$ and library size $l_j$ are random variables on their own distributed as shown:

$$\alpha_i^p \sim \begin{cases} 1, & P = 1 - \delta \\ 1/\epsilon, & P = \delta/2 \\ \epsilon, & P = \delta/2 \end{cases} \quad (4)$$

$$l_j \sim \text{LogNormal}(\mu_l, \sigma_l^2) \quad (5)$$

Here, $\delta$ represents the average probability of perturbing a gene, $\epsilon > 1$ the strength of the multiplicative effect of a perturbation, and $\mu_l, \sigma_l^2$ the mean and variance of the library size scaling factor. All $\delta, \epsilon, \mu_l, \sigma_l^2$ are constant for the entire dataset. Following reasonable priors, we define $\lambda_i = \mu_i^{all} - \mu_i^c$ as the difference between the average perturbed expression and the control expression in the *Norman19* dataset which is also used to estimate $\theta_i, \mu_i^c$, and $\sigma_l^2$. Given this setup, we perform random sampling to generate an array of synthetic datasets from the parameter space in Table 1. Following standard processing, every synthetic dataset is library-size normalized to $10^4$ counts per cell and log1p transformed. We generated $10^4$ synthetic datasets and evaluated Pearson($\Delta$) and MSE on 4 gene sets: (1) all genes, (2) affected genes, which reflects the true simulated perturbed genes ($\alpha_i^p \neq 1$), (3) observed DEGs vs control and (4) observed DEGs vs the rest of the perturbations.

### 3.2. Real Data Experiments

To evaluate the realism of our simulated results, we created analogous experiments in real-world data. We processed and analyzed two datasets commonly used in benchmarks: (1) *Replogle22*, a genome-wide CRISPRi Perturb-seq dataset

(Replogle et al., 2022), and *Norman19*, a CRISPRa Perturb-seq dataset with genes activated alone or in combos of two (Norman et al., 2019). Datasets were randomly downsampled such that each perturbation label had the same number of cells (64 for *Replogle22* and 256 for *Norman19*). For both datasets, we selected the top 8192 highly-variable genes using the `highly_variable_genes` function from the scanpy package (Wolf et al., 2018). We then used the `rank_genes_groups` function from scanpy with the `t-test_overestim_var` method to calculate DEGs with respect to the control cells (DEGs vs Control) and with respect to all other perturbations (DEGs vs Rest). A detailed description of the experiments performed on real data is provided in Appendix A and high-level parameter ranges are available in Table 1.

### 3.3. Proposed Metrics

#### 3.3.1. WEIGHTED DELTA $R^2$: $R_w^2(\Delta)$

Given a set of positive weights $\{w_i\}$ that add to one, average perturbed expression levels $\{\mu_i^{all}\}$, ground truth expression levels $\{\mu_i^p\}$ and pseudobulked predicted values $\{\hat{\mu}_i^p\}$, $i \in \{1, 2, \ldots, g\}$ with $g$ the number of genes in the dataset, we define $R_w^2(\Delta)$ for a single perturbation as follows:

$$R_w^2(\Delta) = 1 - \frac{\sum_i w_i (\Delta_i - \hat{\Delta}_i)^2}{\sum_i w_i (\Delta_i - \bar{\Delta}_w)^2} \quad (6)$$

$$\bar{\Delta}_w = \sum_{i=1}^{g} w_i \Delta_i \quad (7)$$

Where $\Delta_i = \mu_i^p - \mu_i^{all}$ and $\hat{\Delta}_i = \hat{\mu}_i^p - \mu_i^{all}$ represent the real and predicted changes from the average of all perturbed cells respectively. Note that reference values for delta computation $\mu_i^{all}$ are the center of all perturbed cells in the dataset instead of the traditional definition which computes against the control population $\mu_i^c$. We propose $R_w^2(\Delta)$ as a significantly more stringent alternative to Pearson($\Delta$) with the following four advantages. (i) As a goodness of fit metric, the scale and dynamic range of the predictions does matter. It is not enough to estimate the direction of change as with Pearson($\Delta$). (ii) Because we set the reference to the mean of all perturbed cells in the dataset, there is, by definition, no systematic bias from control cells that can inflate metrics unintentionally. (iii) Because of the properties of $R^2$, any constant average predictions ($\hat{\mu}^p = \mu^{all}$) will yield a strictly negative result for any specific perturbation, *i.e.* any positive value automatically means an improvement with respect to the null prediction $\mu^{all}$ (see Appendix B for derivation). (iv) This metric, while still computing in full transcriptomic space, can prioritize more biologically significant genes like DEGs by changing the weights definition.

#### 3.3.2. WEIGHTED ERROR: WMSE

We propose a modified version of the classical MSE regression metric aiming to capture per perturbation nuances. Given a set of weights $\{w_i\}$ that add to one, WMSE is defined as follows:

$$\text{WMSE} = \sum_{i=1}^{g} w_i (\mu_i^p - \hat{\mu}_i^p)^2 \quad (8)$$

While simple, this modification addresses the main pitfalls of error metrics currently used in the task (see 2.3). Unlike standard MSE, WMSE allows for gene signal prioritization, such as for perturbation-specific DEGs. In other words, WMSE is more sensitive to perturbation-specific signals, which is particularly important given that only a small proportion of genes change meaningfully in response to each perturbation (Nadig et al., 2025). Moreover, WMSE can directly replace MSE as training loss for many models allowing for biologically meaningful supervision.

#### 3.3.3. WEIGHTS DEFINITION

Although the weight set $\{w_1, w_2, \ldots, w_g\}$ may be arbitrarily chosen to highlight any biologically relevant signal on the data, here we choose the weights to prioritize perturbation-specific DEGs. This intuitively assigns higher importance to genes that change meaningfully in a perturbation while still allowing consideration of the whole transcriptome. Our weight computation procedure for a single perturbation is the following: (i) we determine $t-$scores for every gene with respect to the rest of the perturbed cells in the dataset (using scanpy's `sc.tl.rank_genes_groups` (Wolf et al., 2018) function with `method="t-test_overestim_var"` and `reference="rest"`) (ii) we apply an absolute value transformation, (iii) we perform min-max normalization to the $[0, 1]$ range, (iv) we square the weights to accentuate differences, and (v) we normalize the whole weight set to add up to 1. The key differentiator of this method is the use of all other perturbed cells as reference for DEG calculation (DEGs vs Rest) instead of the experiment's control (DEGs vs Control). This selection ensures the prioritized genes are the ones that make that perturbation unique from all the others without introducing control bias.

### 3.4. Technical Duplicate Baseline

scRNA-seq perturbation data poses multiple challenges for reproducible perturbation modeling, such as low capture, difficult annotation of perturbed cells, low perturbation efficiency, and high dimensionality which all contribute to high variance in average effect estimation for any given perturbation. To address this problem, we propose a positive performance baseline which tries to answer a simple

*Table 2.* Pearson correlations between metrics and simulation parameters. Comparison between real data experiments and simulation experiments for all the genes. Correlation values lower than $-0.2$ are highlighted in **blue** and Correlation values higher than $0.2$ are highlighted in **red**. Note: Values for the $\epsilon$ parameter on real data are not shown since it can not be disentangled from $\delta$. Other missing values correspond to constant performance over the parameter.

| Metric | Dataset | $\epsilon$ | $g$ | $n_p$ | $\beta$ | $\delta$ | $n_0$ | $k$ | $\mu_l$ |
|---|---|---|---|---|---|---|---|---|---|
| | Norman19 | - | **0.36** | **0.39** | **0.63** | -0.19 | **-0.64** | -0.09 | -0.09 |
| Pearson($\Delta$) | Replogle22 | - | 0.10 | **0.27** | **0.58** | -0.06 | **-0.74** | -0.03 | 0.07 |
| | Simulation | **-0.32** | 0.00 | **0.28** | **0.54** | **-0.28** | **-0.48** | -0.10 | 0.03 |
| | Norman19 | - | **0.21** | **-0.74** | - | **0.70** | - | 0.05 | **-0.42** |
| MSE | Replogle22 | - | 0.17 | **-0.85** | - | **0.63** | - | 0.01 | **-0.58** |
| | Simulation | **0.45** | **-0.44** | **-0.35** | 0.04 | **0.42** | -0.01 | 0.02 | -0.17 |

question: "*how would a technical replicate of the dataset perform in predicting a mean perturbation effect?*". Achieving this level of performance for a model would mean that its prediction errors are comparable to the variance of the experiment itself, defining a performance ceiling. We compute this baseline by randomly dividing the population of cells receiving a perturbation in half and using one half of the cells to predict the other half.

### 3.5. MSE vs WMSE Training

To assess whether WMSE can function not only as an evaluation metric but also as an effective training objective, we retrained GEARS from scratch using default hyperparameters under three loss functions: (i) the original GEARS loss, (ii) the standard unweighted MSE, and (iii) WMSE. On *Replogle22*, we evaluated the unseen gene prediction task, using half of the perturbations for training and validation and reserving the remaining perturbations for testing. On *Norman19*, we evaluated the combination prediction task, training on all single-gene perturbations and half of the two-gene combinations, with the remaining combinations held out for testing. In both settings, we trained two GEARS models to generate predictions for all unseen genes or all held-out combinations, respectively.

To test whether the observed effects generalize beyond GEARS, we additionally evaluated three distinct model architectures on the *Replogle22* task: (i) an MLP baseline ("FMLP") that encodes perturbations using publicly available ESM2 embeddings (Lin et al., 2023), (ii) scLambda (Wang et al., 2024), a variational autoencoder that represents perturbations using LLM-derived text embeddings of gene descriptions, and (iii) scGPT (Cui et al., 2024), a transformer-based foundation model fine-tuned for unseen perturbation prediction. All models were originally trained with an MSE loss to supervise transcriptome-wide expression prediction. For each architecture, we extended the training objective to support WMSE by incorporating perturbation-specific gene weights.

## 4. Experimental Results

### 4.1. *In silico* Screen

The results of our *in silico* simulation study are summarized in Table 2 and Table S1. Across all settings, the number of cells per perturbation ($n_p$) emerged as the strongest determinant of baseline performance for both metrics. This effect is primarily driven by reduced sparsity in the pseudobulk estimates $\mu^p$ as the number of sampled cells increases. Because single-cell expression vectors are highly sparse, increasing $n_p$ alone substantially reduces the distance between $\mu^{all}$ and $\mu^p$ by stabilizing the pseudobulk estimate.

As a reference-based metric, Pearson($\Delta$) was strongly influenced by several dataset properties. In particular, both the number of control cells ($n_0$) and the magnitude of control bias ($\beta$) substantially modulated the apparent performance of the mean baseline. Increased control bias systematically inflated Pearson($\Delta$) scores, consistent with prior observations (Viñas Torné et al., 2025), highlighting the fragility of reference-dependent metrics when the reference population itself is biased. An in-depth analysis of control bias, number of control cells and other parameters influencing Pearson($\Delta$) is reported in Appendices C−E.

For the reference-free MSE, we first observed a reduction in the distance $|\mu^{all} - \mu^p|$ as the mean library size $\mu_l$ increased. This effect likely reflects artifacts introduced by preprocessing: because all datasets are normalized to a fixed library size of $10^4$, cells with larger original library sizes yield smaller normalized expression values overall, leading to reduced distances. More importantly, we observed a consistent degradation in mean-baseline performance as both the probability of perturbing a gene ($\delta$) and the perturbation effect size ($\epsilon$) increased (Fig. 2, Table S1). While statistically expected—stronger perturbations should lie farther from the dataset mean—this result highlights a specific metric artifact: *the mean baseline is a stronger predictor in datasets with weaker perturbations*.

This observation is particularly salient in the context of

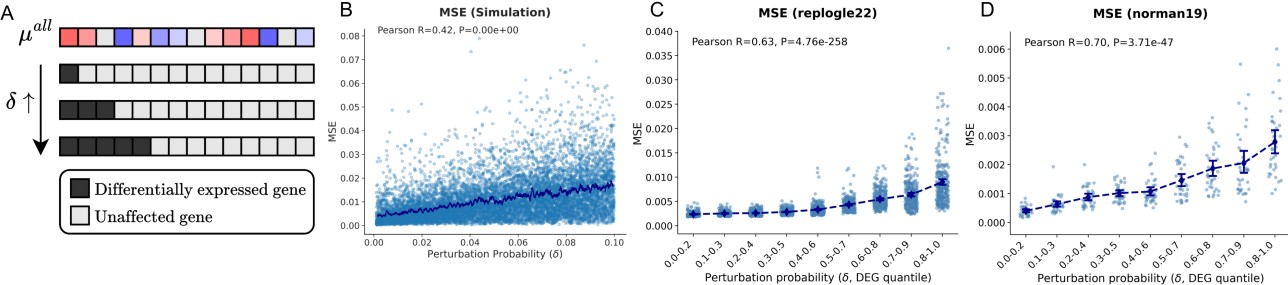

*Figure 2.* (A) Schematic of the simulation sweep in which the average number of perturbed genes increases with the perturbation probability $\delta$. (B) MSE performance of the mean baseline $\mu^{all}$ as a function of the proportion of perturbed genes in simulated data; the trend line denotes a moving average. (C) MSE performance of $\mu^{all}$ as a function of perturbation strength in the *Replogle22* dataset, with perturbations grouped into quantiles based on the number of detected DEGs. (D) Same as (C) for the *Norman19* dataset.

real datasets, where the number of differentially expressed genes varies widely across perturbations. Under our processing pipeline, the median number of DEGs per perturbation was 4 for *Replogle22* and 110 for *Norman19* out of 8192 genes (approximately 0.05% and 1.34%, respectively). Consequently, transcriptome-wide perturbation modeling becomes a *needle-in-a-haystack* regression problem: meaningful signal is confined to a very small subset of genes and is overwhelmed by the vast majority of genes that remain unchanged (Fig. 1b). Under such conditions, it is unsurprising that a degenerate predictor such as the mean baseline ($\mu^{all}$) performs well under global evaluation metrics.

## 4.2. WMSE is Sensitive to Niche Perturbation Effects

By construction, WMSE is designed to address the signal-dilution problem faced in transcriptome-wide perturbation modeling (Fig. 1b). To empirically validate this approach, we compared the sensitivity of standard MSE and WMSE under a controlled perturbation-aware experiment. For each perturbation, we evaluated two candidate predictions against the true perturbation mean $\mu^p$: (i) the mean of all perturbed cells in the dataset ($\mu^{all}$), representing a completely uninformed baseline; and (ii) a modified baseline $\mu^{all*}$ in which the expression values of the top 25 perturbation-specific DEGs (approximately 0.3% of all genes) were replaced with their true values from $\mu^p$. Despite this minimal modification, $\mu^{all*}$ captures the dominant perturbation-specific signal and thus constitutes a robust prediction. While both metrics registered lower error for $\mu^{all*}$ relative to $\mu^{all}$, the magnitude of the improvement differed substantially. Across perturbations, MSE exhibited modest fold reductions of 1.14 and 1.65 for *Replogle22* and *Norman19*, respectively. In contrast, WMSE showed markedly larger fold reductions of 3.26 and 9.65 (Fig. 3), demonstrating substantially greater sensitivity to sparse, perturbation-specific DEG signals. These results indicate that WMSE more effectively distinguishes biologically meaningful improvements over degenerate mean-like predictions.

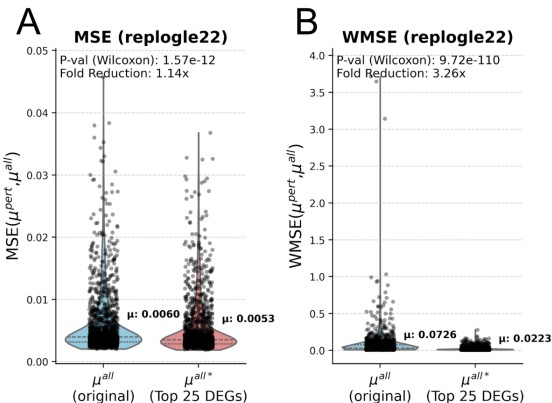

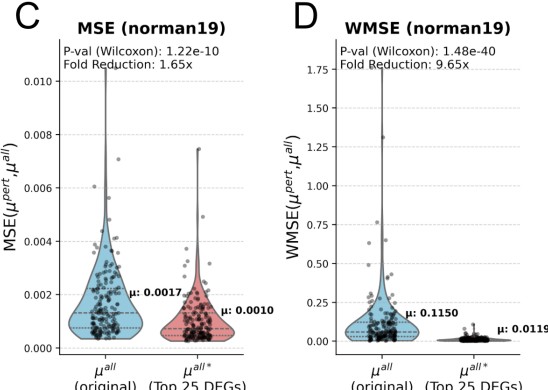

*Figure 3.* Weighted MSE is sensitive to differences in niche perturbation signals. (A) Violin plot showing MSE between true perturbation profiles $\mu^p$ and two possible predictions: the mean baseline $\mu^{all}$ and a modified version $\mu^{all*}$ which sets the top 25 DEGs to be perfectly predicted. (B) Same as (A) but MSE was weighted by the normalized DEG score (WMSE). (C-D) Same as (A-B) but for the *Norman19* dataset.

## 4.3. Weighting Improves Model Metric Sensitivity

As discussed above, commonly used error metrics dilute perturbation-specific DEG signals, systematically favoring uninformative mean-like predictions when perturbations are

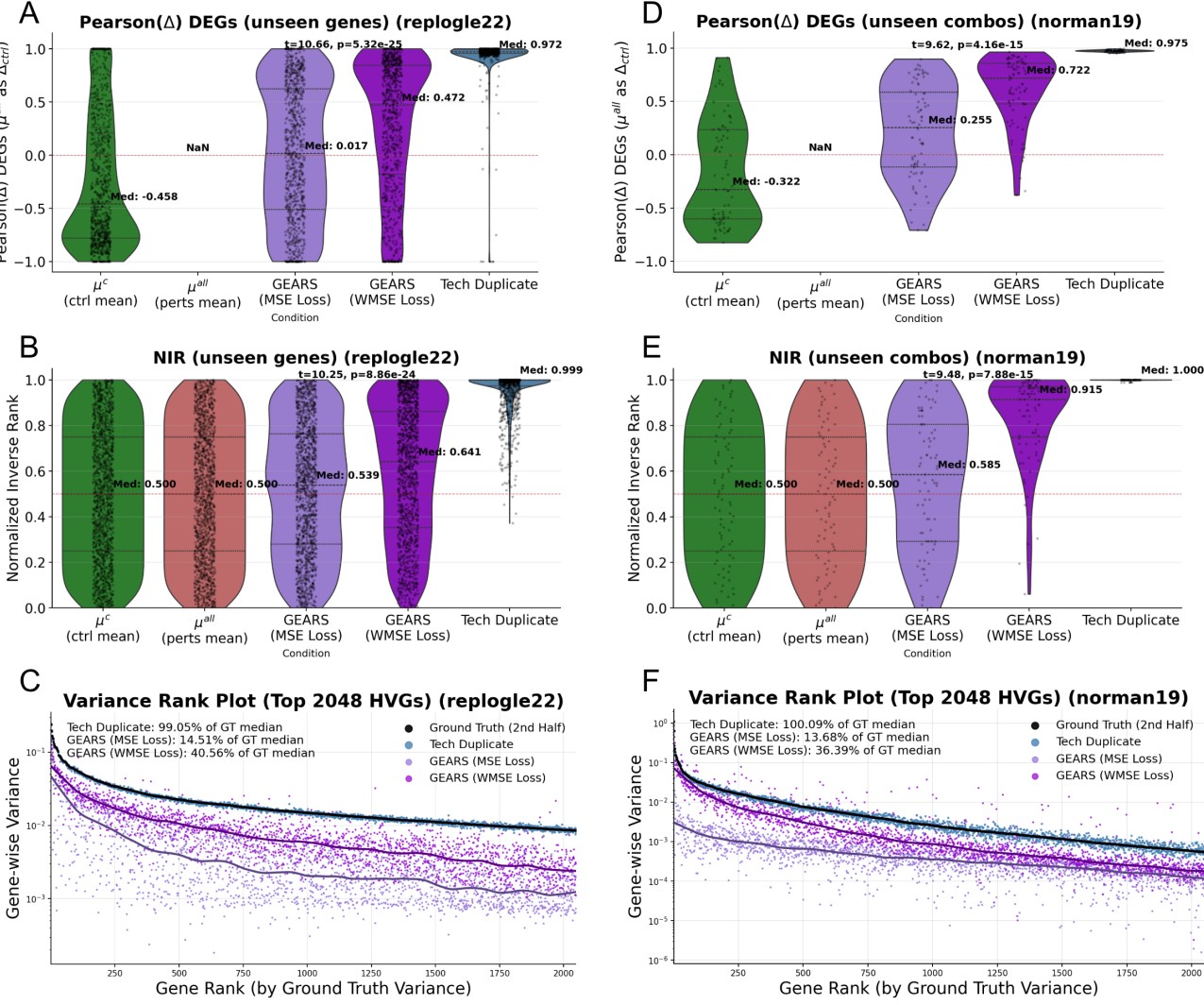

*Figure 4.* DEG score-weighted loss reduces mode collapse and improves model performance. (A) Performance compared between prediction and ground-truth perturbation mean with Pearson($\Delta$) of perturbation-specific DEGs (vs Rest) within the *Replogle22* dataset. X labels: $\mu^c$ (control mean), $\mu^{all}$ (mean of all perturbed cells), predictions from GEARS model with MSE or WMSE loss, and technical duplicate baseline. Means between GEARS MSE/WMSE compared with paired t-test. (B) Same as (A) but for the Normalized Inverse Rank (NIR) metric. (C) Plot showing the top 2048 highly-variable genes ranked by variance in the ground truth pseudobulked dataset. (D-F) Same as (A-C) but for the *Norman19* dataset.

weak (low $\delta$, $\epsilon$) or when control-cell bias ($\beta$) is present. Under these conditions, even models that successfully learn perturbation-specific effects may appear to underperform a degenerate mean baseline ($\mu^{all}$). To address these limitations, we introduce two DEG-weighted evaluation metrics (Section 3.3): WMSE and the DEG score-weighted delta $R^2$, denoted $R_w^2(\Delta)$, in which $\mu^{all}$ serves as the reference for $\Delta$ computation.

In addition, existing benchmarks often lack explicit scale calibration, as positive performance baselines are rarely included. To remedy this, we evaluate three reference predictors: the control mean $\mu^c$, which represents a biased negative baseline; the mean of all perturbed cells $\mu^{all}$, which constitutes an uninformative null baseline; and a technical

duplicate baseline that approximates robust positive performance (Section 3.4).

As shown in Supplemental Fig. S2, our baselines reveal that DEG-based weighting greatly expands the dynamic range of each metric and reinstates a coherent ranking: the biased control prediction performs worst, the mean baseline has null performance, and the technical duplicate baseline establishes the upper bound. Without weighting, this order collapses; in *Replogle22*, for instance, the technical duplicate performs worst on both MSE and $R^2(\Delta)$ despite being the most informative predictor. The apparent anomaly highlights the signal-dilution problem. Technical duplicate estimates are derived from only half the cells in each perturbation (32 in *Replogle22*) and therefore carry more sampling

noise than the population means returned by the baselines. Metrics that treat all genes uniformly reward this noise reduction, even though it is biologically irrelevant. Once genes are re-weighted by their perturbation-specific DEG statistics (WMSE, $R_w^2(\Delta)$), the technical duplicate's superior capture of genuine signal outweighs its higher variance, whereas models that regress toward the dataset mean lose ground. Indeed, for a weak perturbation such as MRPL23, unweighted $\Delta$ vectors from the two technical duplicate halves show virtually no correlation, yet restricting the comparison to the five true DEGs restores a strong correspondence, precisely the behavior our weighting scheme is designed to reward (Supplemental Fig. S3).

### 4.4. MSE vs. WMSE Training

Using the calibrated metrics introduced above, we first evaluated the performance of a well-established perturbation response model, GEARS (Supplemental Fig. S2). GEARS achieved strong performance on the combination prediction task (*Norman19*), which requires extrapolating from single-gene perturbations to two-gene combinations. In contrast, GEARS struggled to outperform the uninformative mean baseline ($\mu^{all}$) on the more challenging unseen gene prediction task (*Replogle22*) (Supplemental Fig. S2). This task involves zero-shot generalization in a dataset characterized by substantially weaker perturbations, with a median of only four DEGs per perturbation.

We hypothesized that this failure mode arises from signal dilution in the optimization objective, as the unweighted MSE loss rewards predictions that resemble the dataset mean. To test this hypothesis, we retrained GEARS on both datasets using either standard MSE or WMSE loss alone and evaluated performance across all metrics (Fig. 4, Supplemental Fig. S4).

As expected, WMSE training significantly improved both WMSE and $R_w^2(\Delta)$ in both datasets (Supplemental Fig. S4). Given the close alignment between the training objective and these evaluation metrics, this result is unsurprising. To ensure that improvements were not driven solely by this alignment, we additionally evaluated performance using two more orthogonal metrics: Pearson($\Delta$) computed after restricting to perturbation-specific DEGs (vs. Rest), and the Normalized Inverse Rank (NIR) (Wu et al., 2026). Neither metric incorporates DEG-based weighting, making them a more independent test of model performance. Additionally, both metrics exhibited robust calibration comparable to WMSE and $R_w^2(\Delta)$ (Supplemental Fig. S2), indicating their suitability for this evaluation.

On the *Replogle22* unseen gene prediction task, WMSE-trained GEARS achieved significant performance gains over MSE-trained models on both Pearson($\Delta$) DEGs ($t = 10.66; p = 5.32 \times 10^{-25}$) and NIR ($t = 10.25; p = $

$8.86 \times 10^{-24}$). These improvements were most pronounced for stronger perturbations with larger numbers of DEGs (Fig. S5). Gene-wise variance profiles further elucidate the source of this effect (Fig. 4c): predictions from MSE-trained GEARS captured only $14.51\%$ of the variance present in the ground-truth pseudobulk data, whereas WMSE training substantially mitigated this collapse, recovering $40.56\%$ of the true perturbation-induced variance. Similar improvements in performance and variance recovery were observed on the *Norman19* dataset (Fig. 4).

To assess whether the benefits of WMSE training generalize beyond GEARS, we evaluated three additional perturbation response architectures on the *Replogle22* task: FMLP, scLambda, and scGPT. These models span diverse architectural paradigms but are all trained with unweighted MSE losses. Consistent with our findings for GEARS, WMSE training improved performance across multiple evaluation metrics, with gains that correlated with perturbation strength (Table S2; Table S3). When evaluation was restricted to the strongest perturbations, WMSE yielded improvements across all metrics for every model.

Taken together, these results demonstrate that DEG-based weighting reshapes the optimization landscape, steering learning away from mean-like solutions and toward sparse, high-variance predictions that better reflect true perturbation effects. The consistency of this behavior across datasets and model architectures suggests that WMSE constitutes a broadly applicable and biologically meaningful training objective for transcriptome-wide perturbation response modeling.

## 5. Conclusion

Recent benchmarks reveal that predicting the perturbed dataset mean often performs much better than expected without any learning taking place and often surpasses fitted model performance on common evaluation metrics. From our analyses on simulated and real-world datasets, we traced this behavior to metric artifacts that reward mode collapse. Our conclusions produced a three-step remedy: (i) adopt DEG-score weighted metrics ($\Delta R_w^2$, WMSE) that elevate predictions which capture niche perturbation signals; (ii) use negative ($\mu^c$), null ($\mu^{all}$), and positive (technical duplicate) baselines to ensure metrics are sufficiently calibrated; and (iii) implement DEG-aware optimization objectives (e.g., WMSE). Under this protocol, the mean baseline falls to null performance and models that capture perturbation-specific effects rise to the top.

## Impact Statement

Accurate *in silico* perturbation response models can shorten drug-discovery cycles, cut laboratory costs, and reduce ani-

mal use by flagging promising candidates before any wet-lab work. Metrics that reward degenerate averages, however, risk elevating brittle models that provide uninformative predictions. By reducing reference bias, implementing calibrated, DEG-aware metrics, and introducing an optimization approach that penalizes mode collapse, our work enables scientists to avoid misleading metric artifacts and steer their resources toward building and evaluating better perturbation response models.

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

## A. Simulation on Real Data

For every parameter in the simulated data, we designed real data experiments as follows:

- $g$: In a $\log_2$ sequence from 2-8192, we randomly downsampled the data to $N$ genes.

- $n_0$: In a $\log_2$ sequence from 1-8192, we randomly downsampled the control cell population to $N$ cells.

- $n_p$: In a $\log_2$ sequence from 2-256 (for *Norman19*) and 2-64 (for *Replogle22*), we randomly selected $N$ cells for each perturbation label.

- $k$: In a $\log_2$ sequence from 1-175 (for *Norman19*) and 1-1334 (for *Replogle22*), we randomly selected $N$ perturbations. We repeated downsampling with 10 random seeds.

- $\beta$: We calculated the $\Delta$ between the mean of all perturbed cells ($\mu^{all}$) and the mean of the control cells ($\mu^c$). We then created synthetic control data by interpolating in equivalent steps of $0.1\Delta$ between $\mu^{all}$ ($0\Delta$) and $\mu^c$ ($1\Delta$), terminating the interpolation at $2\Delta$.

- $\delta$: Because there was no real-data equivalent of this simulated parameter, we mimicked it by downsampling the data to include perturbations with variable numbers of detected DEGs. We first ranked perturbations by the number of significant DEGs detected. We then downsampled the data by selecting perturbations in $20\%$ quantile windows of normalized ranks ($0-0.2, 0.1-0.3, ...$) such that $0-0.2$ had the weakest perturbations and $0.8-1.0$ had the strongest.

- $\epsilon$: We did not find a way of disentangling this parameter from $\delta$ for the real datasets without selecting a different set of genes per perturbations which removes the fairness of the comparison. Consequently, we only simulate for $\delta$.

- $\mu_l$: Within each perturbation, cells were ranked by library size, and the ranks were binned into deciles. Data were downsampled by selecting only cells belonging to each decile in sequence ($0-0.1, 0.1-0.2, ...$).

*Table S1*. Pearson correlations between metric performance of the mean baseline ($\hat{\mu}^p = \mu^{all}$) and dataset parameters in simulation experiments. Correlation values lower than $-0.2$ are highlighted in **blue** and higher than $0.2$ are highlighted in **red**.

| Metric | Gene Group | $\epsilon$ | $g$ | $n_p$ | $\beta$ | $\delta$ | $n_0$ | $k$ | $\mu_l$ |
|---|---|---|---|---|---|---|---|---|---|
| Pearson($\Delta$) ($\uparrow$) | All | **-0.32** | 0.00 | **0.28** | **0.54** | **-0.28** | **-0.48** | -0.10 | 0.03 |
| | Affected | **-0.25** | -0.08 | 0.06 | 0.12 | -0.09 | -0.14 | **-0.68** | -0.01 |
| | DEGs vs Control | **-0.50** | -0.03 | 0.00 | **0.45** | **-0.28** | **-0.39** | **-0.23** | -0.13 |
| | DEGs vs Rest | **-0.33** | -0.04 | **0.20** | **0.35** | 0.00 | **-0.40** | **-0.43** | -0.03 |
| MSE ($\downarrow$) | All | **0.45** | **-0.44** | **-0.35** | 0.04 | **0.42** | -0.01 | 0.02 | -0.17 |
| | Affected | **0.83** | **-0.42** | -0.02 | 0.03 | 0.01 | 0.01 | 0.05 | -0.02 |
| | DEGs vs Control | **0.70** | **-0.46** | **-0.25** | 0.01 | -0.06 | 0.06 | 0.05 | -0.12 |
| | DEGs vs Rest | **0.67** | **-0.48** | **-0.26** | 0.03 | -0.08 | 0.01 | 0.04 | -0.14 |

## B. Upper Bounds of $R^2_w(\Delta)$ Under Constant $\hat{\mu}^p = \mu^{all}$ Predictions

Given the definition of the metric:

$$R^2_w(\Delta) = 1 - \frac{\sum_i w_i (\Delta_i - \hat{\Delta}_i)^2}{\sum_i w_i (\Delta_i - \bar{\Delta}_w)^2} \tag{9}$$

$$\bar{\Delta}_w = \sum_{i=1}^{g} w_i \Delta_i \tag{10}$$

$$\Delta_i = \mu_i^p - \mu_i^{all} \tag{11}$$

$$\hat{\Delta}_i = \hat{\mu}_i^p - \mu_i^{all} \tag{12}$$

Under a constant prediction $\hat{\mu}_i^p = \mu_i^{all}$ any predicted delta becomes $\hat{\Delta}_i = 0$ and the overall metric reduces to:

$$R_w^2(\Delta) = 1 - \frac{\sum_i w_i \Delta_i^2}{\sum_i w_i (\Delta_i - \bar{\Delta}_w)^2} \tag{13}$$

$$\tag{14}$$

Which can be easily shown to be negative by expanding $\sum_i w_i (\Delta_i - \bar{\Delta}_w)^2$ as follows:

$$\sum_i w_i (\Delta_i - \bar{\Delta}_w)^2 = \sum_i w_i \Delta_i^2 - 2\bar{\Delta}_w \sum_i w_i \Delta_i + \bar{\Delta}_w^2 \sum_i w_i \tag{15}$$

Because the set $\{w_i\}$ is normalized to add up to one and under the definition of $\bar{\Delta}_w$ the expression can be reduced to:

$$\sum_i w_i (\Delta_i - \bar{\Delta}_w)^2 = \sum_i w_i \Delta_i^2 - 2\bar{\Delta}_w^2 + \bar{\Delta}_w^2 \tag{16}$$

$$= \sum_i w_i \Delta_i^2 - \bar{\Delta}_w^2 \tag{17}$$

Then, rewriting the original metric value under the expansion we get:

$$R_w^2(\Delta) = 1 - \frac{\sum_i w_i \Delta_i^2}{\sum_i w_i \Delta_i^2 - \bar{\Delta}_w^2} \tag{18}$$

$$\tag{19}$$

From which the rightmost fraction is clearly bounded to be positive (the original fraction was between squared quantities) and greater or equal than 1 making $R_w^2(\Delta) \leq 0$ under the constant prediction case for any perturbation.

## C. Control bias $\beta$

Systematic bias is readily apparent when evaluating DEGs against control. This is illustrated in the *Replogle22* dataset in which a substantial proportion of DEGs are shared across multiple perturbations (Fig. S1a). As detailed in a recent report (Viñas Torné et al., 2025), high performance of the mean baseline on Pearson($\Delta$) can be driven by control bias (Fig. S1b), such that if the effect of any perturbation (compared to the control mean) is similar to the effect of all perturbations compared to control. Under this scenario, the global expression difference ($\Delta^{all} = \mu^{all} - \mu^c$) becomes highly correlated with perturbation-specific differences ($\Delta^p = \mu^p - \mu^c$) as the distinction between perturbed and control states dominates the more subtle differences among perturbations. In our simulations, control bias ($\beta$) showed a high correlation with mean baseline ($\mu^{all}$) performance on Pearson($\Delta$) (Fig. S1c, $r = 0.54$). This behavior was also observed in real data when introducing or removing control bias. For example, with even stronger correlations of $0.58$ and $0.63$ in the *Replogle22* (Fig. S1d) and *Norman19* (Fig. S1e) datasets, respectively.

## D. Number of Control Cells ($n_0$)

- Having observed this control bias, we questioned whether better sampling of the control population might be sufficient to reduce it by better approximating the center of the data (Supplemental Fig. S1f).

- We simulated increasingly higher numbers of control cells and found this reduced the predicting accuracy of the dataset mean (indicating a most robust control) (Supplemental Fig. S1g). However, the simulation also demonstrated that there are diminishing returns from continuing to sample the control population beyond 1000 cells. A similar effect was observed in both the *Norman19* and *Replogle22* dataset (Supplemental Fig. S1h and Supplemental Fig. S1i).

These results highlight that, while greater sampling of the control cell population is sufficient to reduce bias, it cannot eliminate it. Thus, metrics which hinge upon the presence of an unbiased control cell population are fundamentally confounded by these effects. This poses a particular challenge for metrics based on deltas (such as the Pearson($\Delta$)) and DEGs, when deltas and DEGs are calculated with respect to the control cell population. This is a common practice in the field today, as evidenced by the widespread use of these metrics in recent papers (Gong et al., 2023; Istrate et al., 2024; Roohani et al., 2024; Cui et al., 2024; Li et al., 2024a; Wenteler et al., 2025; Tang et al., 2024; Csendes et al., 2025).

## E. Number of perturbations ($k$)

- Another interesting finding of our simulation was a high Pearson($\Delta$) performance of the mean baseline for truly affected genes under a very low number of perturbations (around $0.6$ in the lower $k$ limit on Supplemental Fig. S1k).

- We hypothesize this is explained by the sparsity of the true biological differences when perturbations occur in non-overlapping genes. As exemplified on Supplemental Fig. S1j for a single perturbation, if this perturbation uniquely shows up-regulation of gene 1 and gene 2 and we are under the low $k$ regime, then the mean baseline $\mu^{all}$ will pick up some signal from it and correlating $r(\Delta^p, \Delta^{all})$ will yield a positive results. This behavior is direct consequence of Pearson($\Delta$) focusing on direction changes rather that dynamic range. Note that because of the low probability of gene perturbation in simulation and sparsity of biological signal in the real data, this behavior is the rule rather than the exception.

- Confirming our result and explanation when sub-sampling perturbations in the real data under 10 different seeds we get the same trend when analyzing Pearson ($\Delta$) only on DEGs vs the Rest of perturbations which are a proxy of the real affected genes and are the ones that make every perturbation different from every other (Supplemental Figs. S1l and S1m).

- As the number of perturbations in the dataset increases the probability of overlap between perturbed genes increases while also the pulling effect of a single perturbation on the mean baseline is significantly reduced. In other words $\mu^{all}$ is closer to the origin of the plot in Supplemental Fig. S1j reducing artificial performance inflation of the mean baseline as observed with simulations and real data.

## F. Supplemental figures / tables

*Table S2.* Model Performance Comparison on the *Replogle22* Dataset (mean $\pm$ SEM). **Bolded** values indicate mean performance with WMSE loss is better than base MSE model. Performance on all perturbations ("Unfiltered") and top 10% perturbations by DEGs shown.

| Perturbation Filter | Model | MSE | WMSE | Pearson($\Delta$) | Pearson($\Delta$) DEGs | $R^2(\Delta)$ | $R^2_w(\Delta)$ | NIR |
|---|---|---|---|---|---|---|---|---|
| Unfiltered | FMLP | $0.0082 \pm 0.0003$ | $0.0609 \pm 0.0084$ | $0.1561 \pm 0.0335$ | $0.3413 \pm 0.0673$ | $-0.1468 \pm 0.0491$ | $0.0210 \pm 0.0561$ | $0.6035 \pm 0.0287$ |
| | FMLP (WMSE) | $0.0082 \pm 0.0003$ | $\mathbf{0.0598 \pm 0.0080}$ | $\mathbf{0.1562 \pm 0.0328}$ | $0.2979 \pm 0.0696$ | $-0.1491 \pm 0.0491$ | $\mathbf{0.0235 \pm 0.0556}$ | $0.6000 \pm 0.0290$ |
| | scGPT | $0.0082 \pm 0.0004$ | $0.0661 \pm 0.0103$ | $0.0955 \pm 0.0239$ | $0.2064 \pm 0.0616$ | $-0.0770 \pm 0.0218$ | $0.0447 \pm 0.0356$ | $0.5467 \pm 0.0304$ |
| | scGPT (WMSE) | $0.0095 \pm 0.0003$ | $\mathbf{0.0647 \pm 0.0094}$ | $0.0316 \pm 0.0342$ | $\mathbf{0.2686 \pm 0.0664}$ | $-0.3458 \pm 0.0439$ | $-0.0621 \pm 0.0524$ | $\mathbf{0.5481 \pm 0.0282}$ |
| | scLambda | $0.0071 \pm 0.0004$ | $0.0552 \pm 0.0094$ | $0.2706 \pm 0.0346$ | $0.3833 \pm 0.0730$ | $0.0598 \pm 0.0319$ | $0.2257 \pm 0.0390$ | $0.6740 \pm 0.0305$ |
| | scLambda (WMSE) | $0.0072 \pm 0.0004$ | $\mathbf{0.0533 \pm 0.0092}$ | $\mathbf{0.2716 \pm 0.0330}$ | $\mathbf{0.4495 \pm 0.0692}$ | $0.0353 \pm 0.0334$ | $\mathbf{0.2406 \pm 0.0402}$ | $\mathbf{0.6848 \pm 0.0296}$ |
| | Tech Duplicate | $0.0081 \pm 0.0000$ | $0.0102 \pm 0.0001$ | $0.3785 \pm 0.0058$ | $0.9438 \pm 0.0070$ | $-0.2637 \pm 0.0120$ | $0.7280 \pm 0.0060$ | $0.9617 \pm 0.0023$ |
| | Control Mean | $0.0093 \pm 0.0002$ | $0.1026 \pm 0.0072$ | $0.0634 \pm 0.0114$ | $-0.2527 \pm 0.0212$ | $-0.1081 \pm 0.0083$ | $-0.1150 \pm 0.0109$ | $0.5000 \pm 0.0079$ |
| | Dataset Mean | $0.0080 \pm 0.0001$ | $0.0923 \pm 0.0069$ | N/A | N/A | $-0.0187 \pm 0.0007$ | $-0.0509 \pm 0.0022$ | $0.5000 \pm 0.0079$ |
| 90–100% | FMLP | $0.0103 \pm 0.0012$ | $0.0564 \pm 0.0107$ | $0.5540 \pm 0.0947$ | $0.6878 \pm 0.1219$ | $0.3406 \pm 0.0815$ | $0.4398 \pm 0.1001$ | $0.4973 \pm 0.1234$ |
| | FMLP (WMSE) | $\mathbf{0.0103 \pm 0.0011}$ | $\mathbf{0.0545 \pm 0.0096}$ | $\mathbf{0.5687 \pm 0.0902}$ | $\mathbf{0.7186 \pm 0.1146}$ | $\mathbf{0.3431 \pm 0.0793}$ | $\mathbf{0.4478 \pm 0.0969}$ | $\mathbf{0.5169 \pm 0.1162}$ |
| | scGPT | $0.0151 \pm 0.0018$ | $0.1007 \pm 0.0169$ | $0.2947 \pm 0.0771$ | $0.3896 \pm 0.1112$ | $0.0993 \pm 0.0401$ | $0.1249 \pm 0.0578$ | $0.1364 \pm 0.0569$ |
| | scGPT (WMSE) | $\mathbf{0.0125 \pm 0.0011}$ | $\mathbf{0.0712 \pm 0.0108}$ | $\mathbf{0.5264 \pm 0.0444}$ | $\mathbf{0.7087 \pm 0.0552}$ | $\mathbf{0.2446 \pm 0.0386}$ | $\mathbf{0.3680 \pm 0.0563}$ | $\mathbf{0.3440 \pm 0.1066}$ |
| | scLambda | $0.0120 \pm 0.0021$ | $0.0730 \pm 0.0210$ | $0.2647 \pm 0.1889$ | $0.2776 \pm 0.2368$ | $0.2370 \pm 0.1310$ | $0.2962 \pm 0.1643$ | $0.4947 \pm 0.1445$ |
| | scLambda (WMSE) | $\mathbf{0.0113 \pm 0.0020}$ | $\mathbf{0.0670 \pm 0.0208}$ | $\mathbf{0.3553 \pm 0.1731}$ | $\mathbf{0.3743 \pm 0.2221}$ | $\mathbf{0.2831 \pm 0.1261}$ | $\mathbf{0.3645 \pm 0.1616}$ | $\mathbf{0.5348 \pm 0.1421}$ |
| | Tech Duplicate | $0.0083 \pm 0.0001$ | $0.0123 \pm 0.0004$ | $0.7865 \pm 0.0045$ | $0.9564 \pm 0.0009$ | $0.5622 \pm 0.0097$ | $0.9076 \pm 0.0033$ | $0.9998 \pm 0.0001$ |
| | Control Mean | $0.0280 \pm 0.0007$ | $0.2154 \pm 0.0102$ | $-0.5584 \pm 0.0154$ | $-0.6883 \pm 0.0176$ | $-0.3954 \pm 0.0110$ | $-0.4227 \pm 0.0169$ | $0.0561 \pm 0.0035$ |
| | Dataset Mean | $0.0212 \pm 0.0006$ | $0.1676 \pm 0.0095$ | N/A | N/A | $-0.0416 \pm 0.0044$ | $-0.0711 \pm 0.0076$ | $0.0540 \pm 0.0030$ |

*Table S3.* Model Performance by DEG Quantile on Replogle22 (mean $\pm$ SEM). **Bolded** values indicate mean performance with WMSE loss is better than base MSE model. Model performance broken down by perturbation strength quantile in terms of number of DEGs.

| DEG Quantile | Model | MSE | WMSE | Pearson($\Delta$) | Pearson($\Delta$) DEGs | $R^2(\Delta)$ | $R_w^2(\Delta)$ | NIR |
|---|---|---|---|---|---|---|---|---|
| 0-25% | FMLP | $0.0073 \pm 0.0010$ | $0.0436 \pm 0.0105$ | $0.0642 \pm 0.0848$ | N/A | $-0.4319 \pm 0.1935$ | $-1.0335 \pm 0.9624$ | $0.7426 \pm 0.0673$ |
| | FMLP (WMSE) | $\mathbf{0.0073 \pm 0.0009}$ | $0.0458 \pm 0.0091$ | $0.0269 \pm 0.0783$ | N/A | $-0.4348 \pm 0.1928$ | $-1.1594 \pm 0.9863$ | $\mathbf{0.7445 \pm 0.0681}$ |
| | scGPT | $0.0063 \pm 0.0005$ | $0.0165 \pm 0.0022$ | $-0.0098 \pm 0.0741$ | N/A | $-0.2103 \pm 0.0760$ | $0.1450 \pm 0.3742$ | $0.7567 \pm 0.0696$ |
| | scGPT (WMSE) | $0.0086 \pm 0.0008$ | $0.0266 \pm 0.0027$ | $-0.1265 \pm 0.0780$ | N/A | $-0.6474 \pm 0.1224$ | $-0.2375 \pm 0.3940$ | $0.6507 \pm 0.0821$ |
| | scLambda | $0.0055 \pm 0.0006$ | $0.0237 \pm 0.0117$ | $0.2355 \pm 0.0859$ | N/A | $-0.0429 \pm 0.0947$ | $0.1993 \pm 0.1371$ | $0.8045 \pm 0.0699$ |
| | scLambda (WMSE) | $0.0055 \pm 0.0006$ | $0.0245 \pm 0.0131$ | $0.2274 \pm 0.0819$ | N/A | $-0.0588 \pm 0.0868$ | $0.1894 \pm 0.1810$ | $\mathbf{0.8168 \pm 0.0602}$ |
| | Tech Duplicate | $0.0081 \pm 0.0000$ | $0.0094 \pm 0.0003$ | $0.2116 \pm 0.0056$ | N/A | $-0.6061 \pm 0.0133$ | $0.4055 \pm 0.0381$ | $0.9073 \pm 0.0066$ |
| | Control Mean | $0.0050 \pm 0.0001$ | $0.0180 \pm 0.0037$ | $0.2738 \pm 0.0160$ | N/A | $0.0177 \pm 0.0152$ | $0.1470 \pm 0.0580$ | $0.7261 \pm 0.0120$ |
| | Dataset Mean | $0.0052 \pm 0.0001$ | $0.0206 \pm 0.0029$ | N/A | N/A | $-0.0139 \pm 0.0009$ | $-0.0389 \pm 0.0065$ | $0.7321 \pm 0.0107$ |
| 25-50% | FMLP | $0.0071 \pm 0.0004$ | $0.0538 \pm 0.0230$ | $-0.0008 \pm 0.0496$ | N/A | $-0.2797 \pm 0.0739$ | $-0.1258 \pm 0.0655$ | $0.6724 \pm 0.0388$ |
| | FMLP (WMSE) | $0.0072 \pm 0.0004$ | $\mathbf{0.0516 \pm 0.0206}$ | $\mathbf{0.0164 \pm 0.0474}$ | N/A | $-0.2985 \pm 0.0876$ | $\mathbf{-0.1168 \pm 0.0587}$ | $\mathbf{0.6834 \pm 0.0375}$ |
| | scGPT | $0.0063 \pm 0.0003$ | $0.0496 \pm 0.0271$ | $0.0388 \pm 0.0362$ | N/A | $-0.1313 \pm 0.0456$ | $0.0253 \pm 0.0867$ | $0.7365 \pm 0.0354$ |
| | scGPT (WMSE) | $0.0087 \pm 0.0004$ | $0.0566 \pm 0.0258$ | $-0.1738 \pm 0.0461$ | N/A | $-0.5717 \pm 0.0632$ | $-0.3240 \pm 0.1205$ | $0.6111 \pm 0.0480$ |
| | scLambda | $0.0055 \pm 0.0003$ | $0.0523 \pm 0.0263$ | $0.2463 \pm 0.0505$ | N/A | $0.0198 \pm 0.0568$ | $0.1242 \pm 0.0476$ | $0.7696 \pm 0.0357$ |
| | scLambda (WMSE) | $0.0057 \pm 0.0004$ | $\mathbf{0.0519 \pm 0.0260}$ | $0.2154 \pm 0.0489$ | N/A | $-0.0202 \pm 0.0629$ | $\mathbf{0.1333 \pm 0.0429}$ | $0.7574 \pm 0.0359$ |
| | Tech Duplicate | $0.0080 \pm 0.0000$ | $0.0090 \pm 0.0001$ | $0.2561 \pm 0.0052$ | N/A | $-0.4919 \pm 0.0123$ | $0.5873 \pm 0.0101$ | $0.9494 \pm 0.0049$ |
| | Control Mean | $0.0053 \pm 0.0001$ | $0.0423 \pm 0.0054$ | $0.2710 \pm 0.0158$ | N/A | $0.0215 \pm 0.0146$ | $0.1091 \pm 0.0163$ | $0.6609 \pm 0.0108$ |
| | Dataset Mean | $0.0055 \pm 0.0000$ | $0.0443 \pm 0.0051$ | N/A | N/A | $-0.0121 \pm 0.0008$ | $-0.0332 \pm 0.0024$ | $0.6549 \pm 0.0106$ |
| 50-75% | FMLP | $0.0072 \pm 0.0004$ | $0.0595 \pm 0.0143$ | $0.1430 \pm 0.0524$ | $0.3248 \pm 0.0853$ | $-0.1127 \pm 0.0652$ | $0.1113 \pm 0.0693$ | $0.6458 \pm 0.0439$ |
| | FMLP (WMSE) | $0.0072 \pm 0.0005$ | $\mathbf{0.0593 \pm 0.0145}$ | $\mathbf{0.1462 \pm 0.0522}$ | $\mathbf{0.3517 \pm 0.0868}$ | $-0.1151 \pm 0.0634$ | $0.1110 \pm 0.0700$ | $0.6455 \pm 0.0450$ |
| | scGPT | $0.0068 \pm 0.0003$ | $0.0627 \pm 0.0170$ | $0.0960 \pm 0.0363$ | $0.2311 \pm 0.0822$ | $-0.0620 \pm 0.0271$ | $0.0590 \pm 0.0512$ | $0.6173 \pm 0.0425$ |
| | scGPT (WMSE) | $0.0088 \pm 0.0003$ | $0.0679 \pm 0.0153$ | $-0.0307 \pm 0.0517$ | $0.2305 \pm 0.0910$ | $-0.3917 \pm 0.0632$ | $-0.0807 \pm 0.0690$ | $\mathbf{0.6176 \pm 0.0424}$ |
| | scLambda | $0.0060 \pm 0.0003$ | $0.0502 \pm 0.0149$ | $0.3025 \pm 0.0486$ | $0.5103 \pm 0.0873$ | $0.0714 \pm 0.0396$ | $0.2955 \pm 0.0611$ | $0.7004 \pm 0.0459$ |
| | scLambda (WMSE) | $0.0063 \pm 0.0004$ | $\mathbf{0.0488 \pm 0.0147}$ | $0.2876 \pm 0.0488$ | $\mathbf{0.5200 \pm 0.0908}$ | $0.0212 \pm 0.0470$ | $\mathbf{0.3072 \pm 0.0654}$ | $\mathbf{0.7184 \pm 0.0459}$ |
| | Tech Duplicate | $0.0080 \pm 0.0000$ | $0.0101 \pm 0.0003$ | $0.3688 \pm 0.0059$ | $0.9598 \pm 0.0050$ | $-0.2986 \pm 0.0138$ | $0.7707 \pm 0.0074$ | $0.9905 \pm 0.0019$ |
| | Control Mean | $0.0072 \pm 0.0002$ | $0.1153 \pm 0.0198$ | $0.1077 \pm 0.0180$ | $-0.1502 \pm 0.0277$ | $-0.1061 \pm 0.0152$ | $-0.0983 \pm 0.0143$ | $0.4675 \pm 0.0100$ |
| | Dataset Mean | $0.0065 \pm 0.0001$ | $0.1116 \pm 0.0193$ | N/A | N/A | $-0.0159 \pm 0.0012$ | $-0.0547 \pm 0.0043$ | $0.4786 \pm 0.0110$ |
| 75-90% | FMLP | $0.0109 \pm 0.0008$ | $0.0766 \pm 0.0116$ | $0.2204 \pm 0.0803$ | $0.3589 \pm 0.1155$ | $-0.0818 \pm 0.1018$ | $-0.0197 \pm 0.1186$ | $0.4000 \pm 0.0675$ |
| | FMLP (WMSE) | $\mathbf{0.0108 \pm 0.0007}$ | $\mathbf{0.0757 \pm 0.0111}$ | $0.2163 \pm 0.0779$ | $0.3346 \pm 0.1176$ | $\mathbf{-0.0663 \pm 0.0864}$ | $\mathbf{-0.0039 \pm 0.1057}$ | $0.3574 \pm 0.0649$ |
| | scGPT | $0.0106 \pm 0.0004$ | $0.0797 \pm 0.0160$ | $0.1374 \pm 0.0582$ | $0.1532 \pm 0.0928$ | $-0.0261 \pm 0.0453$ | $-0.0139 \pm 0.0723$ | $0.2637 \pm 0.0376$ |
| | scGPT (WMSE) | $0.0108 \pm 0.0004$ | $\mathbf{0.0716 \pm 0.0150}$ | $\mathbf{0.2322 \pm 0.0674}$ | $\mathbf{0.3432 \pm 0.1073}$ | $-0.0846 \pm 0.0793$ | $\mathbf{0.0716 \pm 0.1000}$ | $\mathbf{0.3912 \pm 0.0524}$ |
| | scLambda | $0.0094 \pm 0.0004$ | $0.0617 \pm 0.0107$ | $0.2799 \pm 0.0894$ | $0.4445 \pm 0.1164$ | $0.0737 \pm 0.0852$ | $0.2010 \pm 0.0986$ | $0.5113 \pm 0.0750$ |
| | scLambda (WMSE) | $0.0095 \pm 0.0009$ | $\mathbf{0.0591 \pm 0.0092}$ | $\mathbf{0.3031 \pm 0.0833}$ | $\mathbf{0.4808 \pm 0.1073}$ | $0.0636 \pm 0.0872$ | $\mathbf{0.2023 \pm 0.1023}$ | $\mathbf{0.5211 \pm 0.0743}$ |
| | Tech Duplicate | $0.0081 \pm 0.0001$ | $0.0112 \pm 0.0002$ | $0.6035 \pm 0.0056$ | $0.9627 \pm 0.0009$ | $0.1919 \pm 0.0128$ | $0.8308 \pm 0.0056$ | $0.9997 \pm 0.0001$ |
| | Control Mean | $0.0141 \pm 0.0003$ | $0.1222 \pm 0.0096$ | $-0.2905 \pm 0.0258$ | $-0.4077 \pm 0.0360$ | $-0.3446 \pm 0.0181$ | $-0.3587 \pm 0.0230$ | $0.2065 \pm 0.0070$ |
| | Dataset Mean | $0.0106 \pm 0.0002$ | $0.1028 \pm 0.0095$ | N/A | N/A | $-0.0272 \pm 0.0022$ | $-0.0626 \pm 0.0057$ | $0.1894 \pm 0.0048$ |
| 90-100% | FMLP | $0.0103 \pm 0.0012$ | $0.0564 \pm 0.0107$ | $0.5540 \pm 0.0947$ | $0.6878 \pm 0.1219$ | $0.3406 \pm 0.0815$ | $0.4398 \pm 0.1001$ | $0.4973 \pm 0.1234$ |
| | FMLP (WMSE) | $0.0103 \pm 0.0011$ | $\mathbf{0.0545 \pm 0.0096}$ | $\mathbf{0.5687 \pm 0.0902}$ | $\mathbf{0.7186 \pm 0.1146}$ | $0.3431 \pm 0.0793$ | $\mathbf{0.4478 \pm 0.0969}$ | $\mathbf{0.5169 \pm 0.1162}$ |
| | scGPT | $0.0151 \pm 0.0018$ | $0.1007 \pm 0.0169$ | $0.2947 \pm 0.0771$ | $0.3896 \pm 0.1112$ | $0.0993 \pm 0.0401$ | $0.1249 \pm 0.0578$ | $0.1364 \pm 0.0569$ |
| | scGPT (WMSE) | $\mathbf{0.0125 \pm 0.0011}$ | $\mathbf{0.0712 \pm 0.0108}$ | $\mathbf{0.5264 \pm 0.0444}$ | $\mathbf{0.7087 \pm 0.0552}$ | $\mathbf{0.2446 \pm 0.0386}$ | $\mathbf{0.3680 \pm 0.0563}$ | $\mathbf{0.3440 \pm 0.1066}$ |
| | scLambda | $0.0120 \pm 0.0021$ | $0.0730 \pm 0.0210$ | $0.2647 \pm 0.1889$ | $0.2776 \pm 0.2368$ | $0.2370 \pm 0.1310$ | $0.2962 \pm 0.1643$ | $0.4947 \pm 0.1445$ |
| | scLambda (WMSE) | $\mathbf{0.0113 \pm 0.0020}$ | $\mathbf{0.0670 \pm 0.0208}$ | $\mathbf{0.3553 \pm 0.1731}$ | $\mathbf{0.3743 \pm 0.2221}$ | $\mathbf{0.2831 \pm 0.1261}$ | $\mathbf{0.3645 \pm 0.1616}$ | $\mathbf{0.5348 \pm 0.1421}$ |
| | Tech Duplicate | $0.0083 \pm 0.0001$ | $0.0123 \pm 0.0004$ | $0.7865 \pm 0.0045$ | $0.9564 \pm 0.0009$ | $0.5622 \pm 0.0097$ | $0.9076 \pm 0.0033$ | $0.9998 \pm 0.0001$ |
| | Control Mean | $0.0280 \pm 0.0007$ | $0.2154 \pm 0.0102$ | $-0.5584 \pm 0.0154$ | $-0.6883 \pm 0.0176$ | $-0.3954 \pm 0.0110$ | $-0.4227 \pm 0.0169$ | $0.0561 \pm 0.0035$ |
| | Dataset Mean | $0.0212 \pm 0.0006$ | $0.1676 \pm 0.0095$ | N/A | N/A | $-0.0416 \pm 0.0044$ | $-0.0711 \pm 0.0076$ | $0.0540 \pm 0.0030$ |

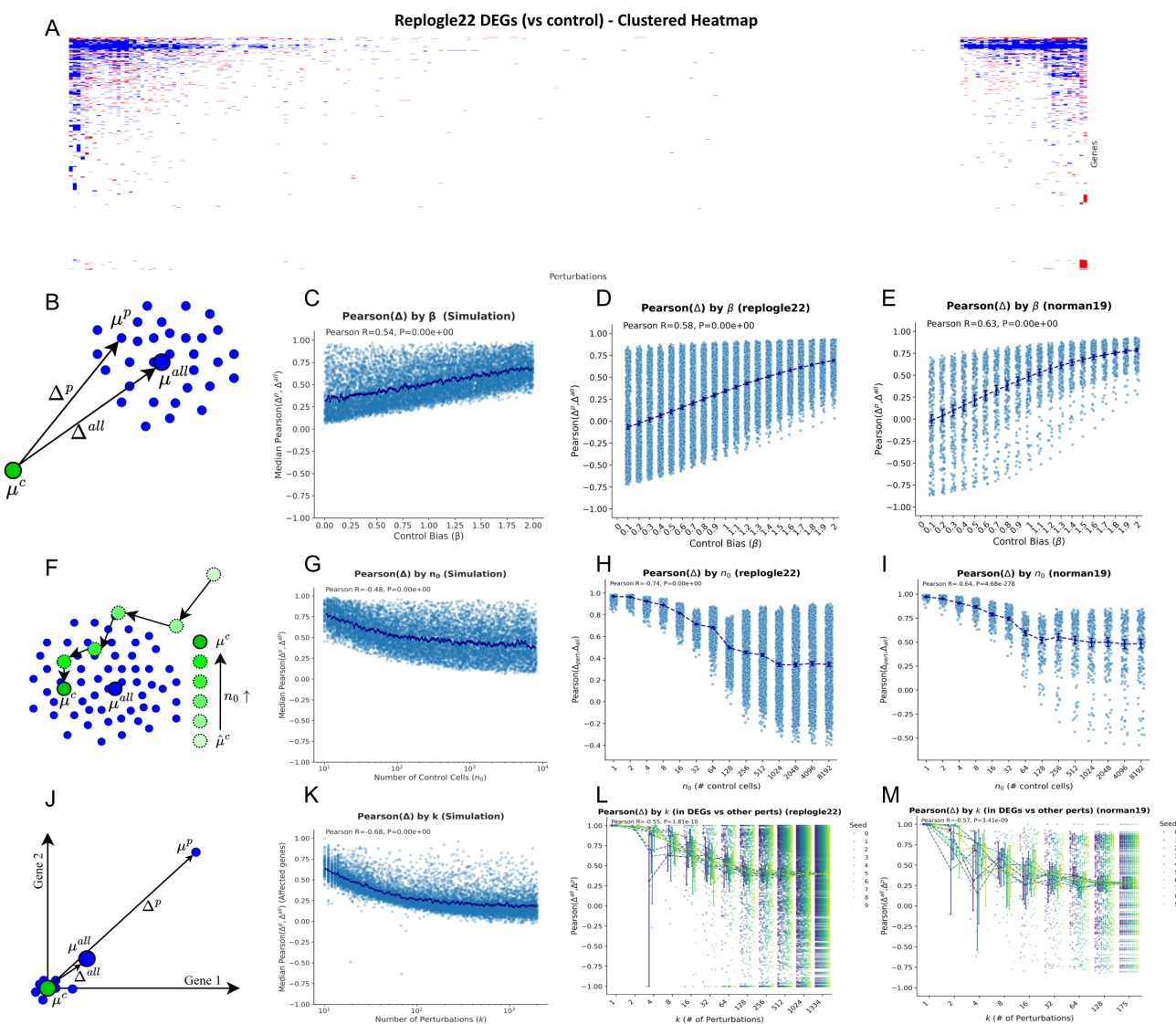

*Figure S1.* Supplemental figure accompanying Fig. 2. (A) Trinary (up, down, or unchanged) clustermap of significant differentially expressed genes of every perturbation against the control population for the *Replogle22* dataset. Genes and perturbations were downsampled randomly (to 256 and 2048 respectively) due to restrictions in plotting software. (B) Schematic showing high correlation between $\Delta^p$ and $\Delta^{all}$ due to the systematic bias of $\mu^c$. (C) Pearson($\Delta$) performance of the mean baseline ($\mu^{all}$) under increasing control bias ($\beta$) in simulations. Trend line shows moving average. (D) Pearson($\Delta$) performance of $\mu^{all}$ under increasing $\beta$ in the *Replogle22* dataset ($\beta = 0$ no control bias, $\beta = 1$ exact dataset control bias, $\beta = 2$ double the dataset control bias). Trend line shows mean and 95% CI of the mean. (E) Same as (D) in the *Norman19* dataset. (F) Diagram showing the effect of increasing the number of control cells ($n_0$) on improving estimation of the control mean ($\mu^c$) also reducing systematic bias. (G) Plot showing the effect of increasing control cell number ($n_0$) on Pearson($\Delta^p, \Delta^{all}$), which is the similarity between $\mu^p - \mu^c$ and $\mu^{all} - \mu^c$, in simulated data. (H-I) Same as (G), but in real datasets *Replogle22* and *Norman19* respectively. (J) Plot illustrating the biasing of $\mu^{all}$ by a strong perturbation $\mu^p$. The effect of this bias is to increase the similarity in direction between $\Delta^p$ and $\Delta^{all}$, especially when the dataset contains fewer perturbations in the first place to moderate this single-perturbation influence. (K) Plot showing the effect of number of perturbations in a simulated dataset ($k$) on the Pearson($\Delta^p, \Delta^{all}$). (L-M), same as (K) except in real datasets and with 10 random seeds for selection of $k$ perturbations.

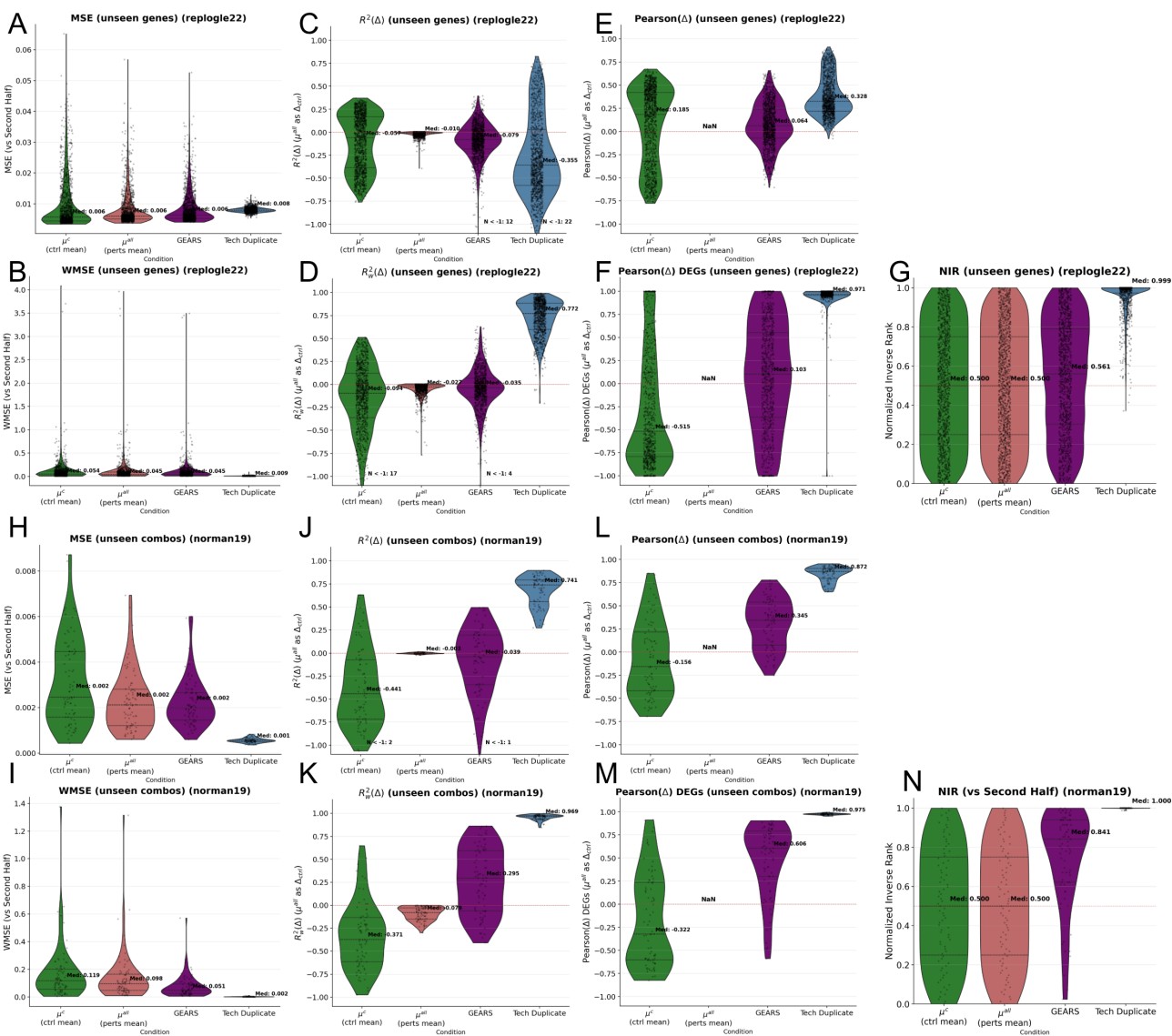

*Figure S2.* DEG-aware metrics, elimination of control bias, and addition of negative and positive baselines provide greater sensitivity and calibration to assess perturbation response model performance. (A) In the *Replogle22* dataset, MSE between ground truth and baselines or model predictions. $\mu^c$ (control cell mean), $\mu^{all}$, GEARS model predictions, and technical duplicate baseline are shown. (B) Same as (A) but measuring error with DEG score-weighted MSE (WMSE) instead of MSE. (C-D) Same as (A-B) but using $R^2(\Delta^p, \hat{\Delta}^p)$ and DEG score-weighted $R^2_w(\Delta^p, \hat{\Delta}^p)$ as the error metric, where $\hat{\Delta}^p$ is $\hat{\mu}^p - \mu^{all}$ and $\Delta^p$ is $\mu^p - \mu^{all}$. (E-F) Same as (C-D) except with Pearson instead of $R^2$ and filtering for DEGs (per perturbation) instead of weighting by DEG score. (G) Same as (A-F) but for the normalized inverse rank metric. (H-N) Same as (A-G) but for the *Norman19* dataset in which the task was prediction of unseen gene combos.

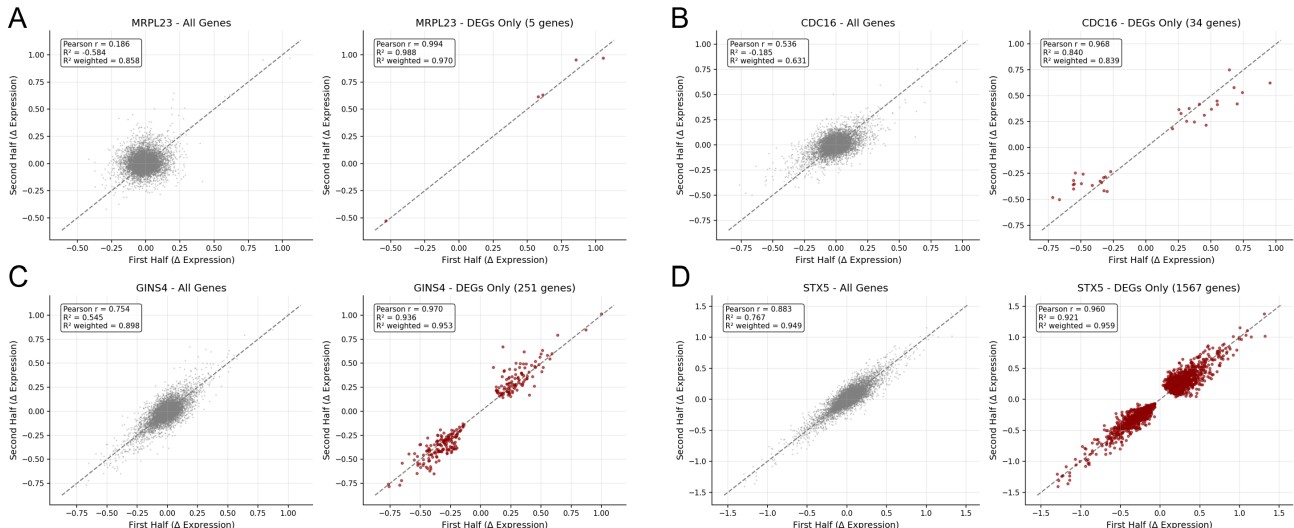

*Figure S3.* Effect of perturbation strength (measured by number of significant DEGs) on $R^2_w(\Delta^p, \hat{\Delta}^p)$ metric in technical duplicate baseline. (A) Scatter plots showing correlation between perturbation effects ($\Delta = \mu^p - \mu^{all}$) when using the first half of the data to predict the second half for MRPL23 with and without filtering for only MRPL23-specific DEGs. DEG-weighted and regular $R^2$ shown, along with Pearson correlation. (B) Same as (A) for CDC16. (C) Same as (A) for GINS4. (D) Same as (A) for STX5.

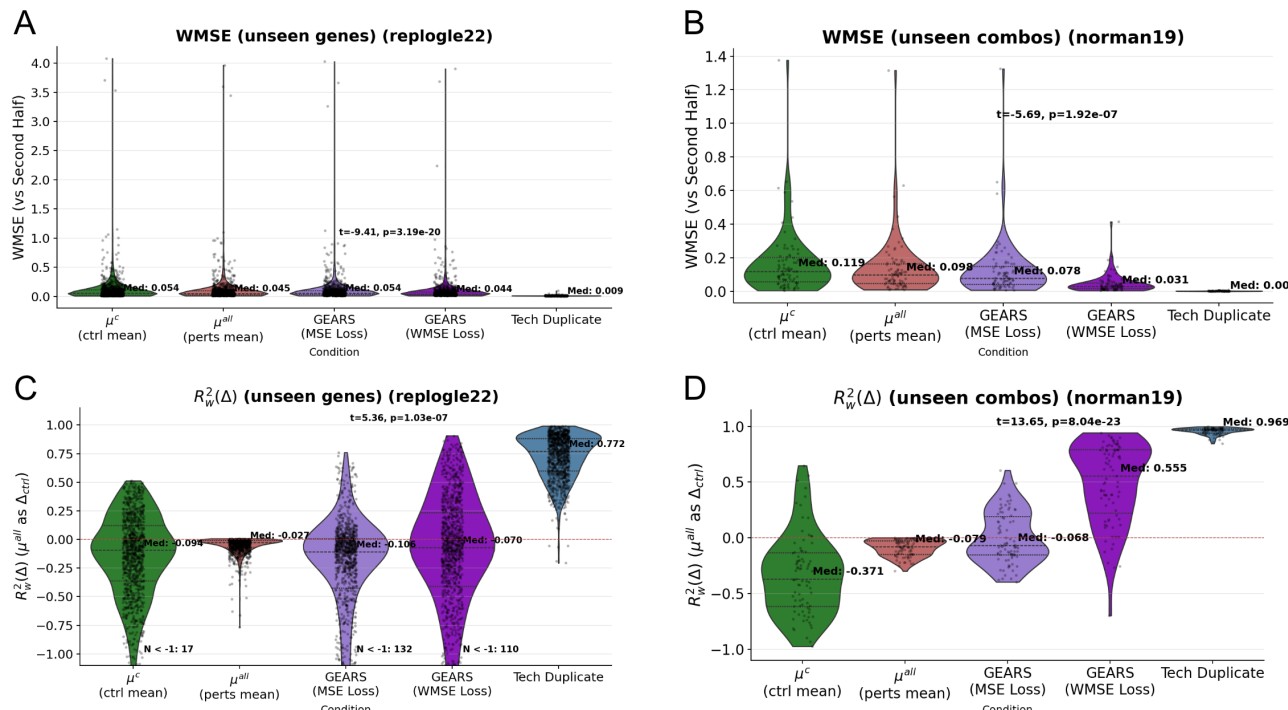

*Figure S4.* Supplemental figure accompanying Fig. 4. A) In the *Replogle22* dataset, WMSE between prediction and ground-truth perturbation mean. X labels: $\mu^c$ (control mean), $\mu^{all}$ (mean of all perturbed cells), predictions from GEARS model with MSE or WMSE loss, and technical duplicate baseline. Means between GEARS MSE/WMSE compared with paired t-test. (B) Same as (A) but for the *Norman19* dataset. (C-D) Same as (A-B) but for $R^2_w(\hat{\Delta}^p, \Delta^p)$, the DEG score-weighted $R^2$ between predicted ($\hat{\Delta}^p$) vs ground-truth perturbation effect ($\Delta^p$). For $\Delta$ calculations, $\mu^{all}$ is the reference.

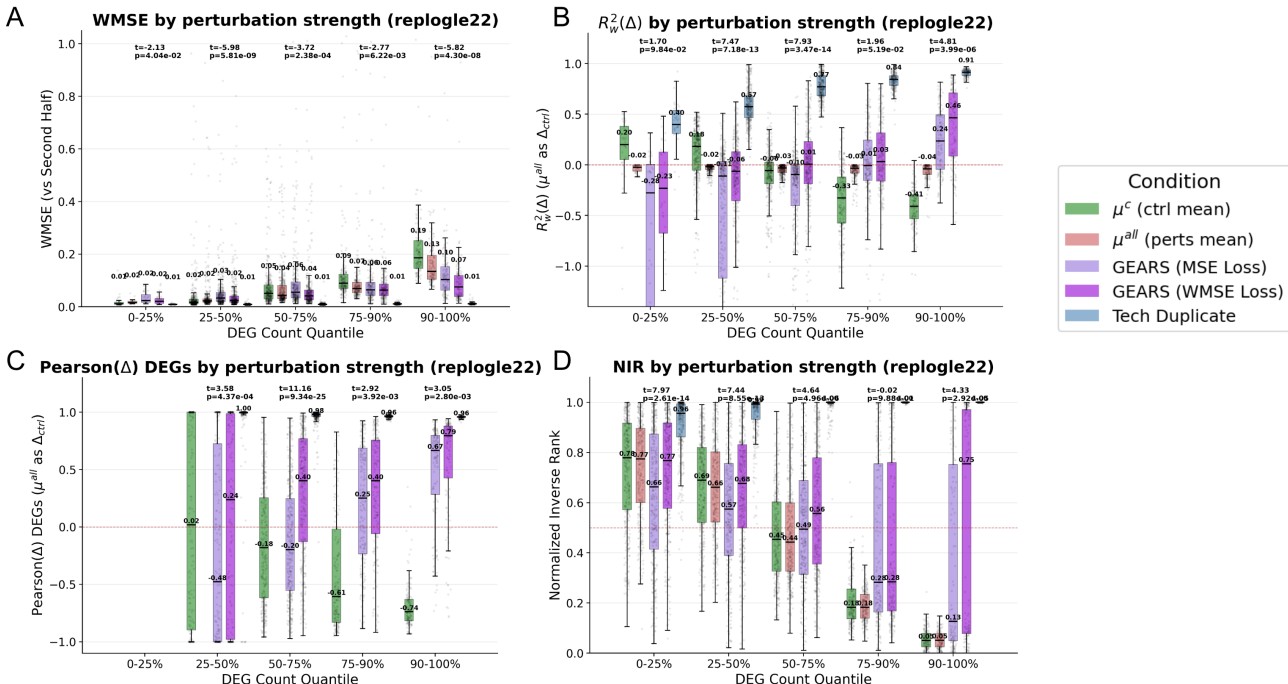

*Figure S5.* DEG score-weighted loss improves model performance on unseen gene prediction task, especially for stronger perturbations. (A) Performance of baselines and GEARS perturbation prediction (with MSE or WMSE loss), compared using WMSE metric vs ground truth perturbation mean ($\mu^p$), grouped by the quantile range of perturbations tested (quantile ranges based on number of DEGs for each perturbation). Paired t-test conducted for each quantile range between GEARS with MSE vs GEARS with WMSE loss, with t-score and p value shown on plot. Median of each prediction within each quantile range also shown. (B) Same as (A) but for $R^2_w(\Delta^p, \hat{\Delta}^p)$ (DEG score-weighted R2 between predicted vs ground-truth perturbation effect). (C) Same as (B) but for Pearson correlation with data filtered to only include perturbation-specific DEGs (vs Rest). Note that the 0-25% quantile is missing because there were no DEGs for perturbations in this quantile. (D) Same as (C) but for the Normalized Inverse Rank (NIR) metric.

