# OpenReview forum: "Needles in the Haystack: Addressing Signal Dilution Improves scRNA-seq Perturbation Response Modeling and Evaluation"
_ICML.cc/2026/Conference — ICML 2026 regular_

### Official Review · Reviewer_2Pb8 · 2026-03-08

**Soundness:** 3
**Presentation:** 4
**Significance:** 4
**Originality:** 3
**Overall Recommendation:** 4
**Confidence:** 4

**Summary:**

This paper investigates a critical anomaly in recent single-cell RNA sequencing (scRNA-seq) perturbation modeling benchmarks: why do simple mean-predicting baselines frequently outperform sophisticated deep learning and foundation models? Through rigorous in silico simulations and the analysis of two real-world datasets (Replogle22 and Norman19), the authors trace this paradox to "signal dilution". Because true genetic perturbation effects are extremely sparse (altering only a tiny fraction of the transcriptome) standard unweighted evaluation metrics like Mean Squared Error (MSE) and Pearson correlation systematically reward models that conservatively predict the dataset mean.

To resolve this, the authors introduce differentially expressed gene (DEG)-aware metrics: Weighted MSE (WMSE) and weighted delta $R^2$ ($R^2_w(\Delta)$). They also propose a calibrated evaluation framework using control mean (negative baseline), dataset mean (null baseline), and a "technical duplicate" (positive upper bound). Finally, they demonstrate that using WMSE as a drop-in replacement training loss substantially reduces mode collapse and improves the zero-shot prediction capabilities of several established models, including GEARS and scGPT.

**Compliance With Llm Reviewing Policy:**

Affirmed.

**Final Justification:**

My core concern is that results and evaluation is based on two datasets. This concern was not adressed nor discuss, so I am keeping my current score as it is

**Key Questions For Authors:**

- Global Reconstruction vs. Niche Signal: You showed that WMSE training recovers higher variance for DEGs. Does this objective sacrifice the model's ability to accurately reconstruct the non-DEG (background) genes, or does performance on unaffected genes remain stable?

- Hyperparameter Sensitivity: In your weight computation, you apply an absolute value transformation, min-max normalization, and then square the weights to accentuate differences. How sensitive is the final model performance to this specific pipeline? Would a simpler softmax or linear scaling work similarly?

**Limitations:**

Yes. The authors appropriately discuss the limitations surrounding the difficulty of modeling sparse signals and the inherent noise in single-cell data. However, they could expand slightly on the potential risks of forcing models to focus exclusively on highly variable DEGs, which might cause models to ignore subtle but biologically relevant pathway-level shifts that do not pass strict significance thresholds.

**Strengths And Weaknesses:**

## Strengths :

- Soundness : The methodology is very rigorous and effectively diagnoses a pervasive benchmarking flaw. Using controlled simulations to isolate the impact of variables like control bias, perturbation strength, and sampling size before validating those findings on real-world data (Replogle22 and Norman19) is a gold-standard approach.

- Presentation : The paper is well structured. The narrative flows intuitively from identifying the problem via simulations to proposing metric solutions, and finally demonstrating those solutions as an optimization objective.

- Significance: This work is significant. The ML-for-biology community has been bottlenecked by evaluations that inadvertently reward uninformative predictions. By exposing the flaws in unweighted metric evaluations and providing a drop-in loss function that fixes the optimization landscape, this paper will likely influence how all future single-cell foundation models are trained and evaluated.

- Originality: While loss re-weighting is a standard technique in machine learning, its specific formulation here using DEGs against all other perturbations to strip away control bias is a highly original and domain-aware solution to the signal dilution problem. The combination of newly calibrated baselines and targeted metrics provides a fresh, much-needed perspective on perturbation modeling.

## Weaknesses :


- Main weakness is that results and evaluation is based on two datasets. Extending the evaluation to more datasets would massively strengthen the contribution of the paper, as it would prove the generalization of its conclusions.
- The introduction of the "technical duplicate" baseline (splitting the target perturbation cells in half to predict one from the other) is a good and necessary method for establishing a realistic performance ceiling given the inherent noise of scRNA-seq data. This was already introduced however already in prior work in : TxPert: Leveraging Biochemical Relationships for Out-of-Distribution Transcriptomic Perturbation Prediction https://arxiv.org/abs/2505.14919 , I believe this deserves reframing and further discussion in context of the missing reference.
- A minor weakness is that the weighting scheme inherently assumes that DEGs are the only biological signals that matter, potentially ignoring more complex, subtle transcriptomic shifts.

---

> ### Author Rebuttal · Authors · 2026-03-31
>
> We thank the reviewer for recognizing the merits of our work and their appreciation of our methodological steps in benchmarking. We address their main concerns as follows:
>
> **Global Reconstruction vs. Niche Signal:** Background gene reconstruction remains effectively unchanged under WMSE training. This is directly evidenced in the MSE column for weak perturbations in Table S3 (0-50% bins), where all perturbations have ≤4 DEGs out of 8,192 genes, meaning MSE is almost exclusively measuring background gene expression prediction fidelity. With the exception of scGPT, MSE values are statistically indistinguishable between MSE and WMSE trained models. This holds for stronger perturbations (50-100% bins) and aggregated performance as well (Table S2).
>
> **Hyperparameter Sensitivity:** We recognize the importance of examining alternative weight definitions. An ablation study on Norman19 using NIR (a weight-free calibrated metric) showed our current WMSE loss weighting scheme outperforms four alternatives: (1) direct squaring, (2) absolute value with linear mapping, (3) absolute value with softmax, and (4) absolute value with normalized ranking. We will include these ablations and accompanying discussion in the final manuscript.
>
> **Subtle pathway-level shifts:** Our WMSE loss employs continuous weights across all gene features without imposing any filters on the DEGs. Consequently, even weakly-perturbed genes contribute to the overall loss. The weighting scheme (DEG test score values) is similar to a ranking scheme frequently employed for preranked GSEA, a method used to assess pathway-level impacts of perturbations in DGE analysis. Consequently, this approach should be adequate for capturing pathway-level changes.
>
> **Technical duplicate:** We thank the reviewer for highlighting TxPert, which we agree merits discussion as an unpublished preprint relevant to our work. In comparing our work and TxPert, we find that there are two key differences between the baselines in these studies. The first is conceptual: TxPert defines experimental reproducibility through repeated sampling, whereas we perform a single split and treat the resulting technical duplicate as a held-out prediction target. Our formulation is arguably closer to real-world scenarios in which researchers might have access to technical replicates (additional cells drawn from the same sequenced samples). The second is practical: repeated sampling renders DEG-based weight computation (for WMSE) and whole-space metrics that scale quadratically with the number of perturbations (e.g., NIR) computationally intractable. We will include a citation and appropriate discussion of TxPert in the final manuscript.
>
> **Focus on DEGs:** We fully agree that DEGs are not the only biologically meaningful signals, but they are among the most salient and directly addressable. Importantly, the WMSE framework is not limited to DEG-derived weights: weights can be defined to emphasize any first-order signal of interest (e.g., specific pathways, transcription factors, or custom gene sets). That said, higher-order interactions remain beyond the scope of this loss formulation and represent a natural direction for future work.

---

> > ### Author Rebuttal · Reviewer_2Pb8 · 2026-04-01
> >
> > My core concern is that results and evaluation is based on two datasets. This concern was not adressed nor discuss, so I am keeping my current score as it is

---

> > > ### Author Response · Authors · 2026-04-03
> > >
> > > We thank the reviewer for raising this point, which we should have addressed directly in our initial rebuttal. Replogle22 and Norman19 were selected because they are the most common benchmarking datasets used in the studies that motivated this work, and they are complementary in important ways: Replogle22 is a large-scale single-gene CRISPRi screen with sparse effects, while Norman19 is a smaller combinatorial CRISPRa screen with stronger perturbation signatures. Beyond these two real-world datasets, our simulation study (10,000 synthetic datasets spanning a broad parameter space) provides further evidence that the signal dilution phenomenon and additional metric artifacts generalize across a range of dataset parameters &mdash; though this was used to support our diagnostic claims, not for model training. The consistency of our findings across these distinct experimental designs provides evidence of generalizability, however we agree that additional datasets would further strengthen this evidence. We will discuss extended dataset coverage as a limitation and direction for future work in the final manuscript.

---

### Official Review · Reviewer_r9z5 · 2026-03-11

**Soundness:** 3
**Presentation:** 3
**Significance:** 3
**Originality:** 4
**Overall Recommendation:** 5
**Confidence:** 5

**Summary:**

This paper investigates why simple mean baselines often outperform deep learning models in single-cell perturbation response prediction. Through large-scale simulations (10,000 synthetic datasets) and analysis of two real-world Perturb-seq datasets (Replogle22 and Norman19), the authors identify "signal dilution" as a key driver: perturbation effects are confined to a small fraction of genes, causing unweighted metrics like MSE to reward mean-like predictions. A central concept considered by the manuscript is the introduction of DEG-aware evaluation metrics (WMSE, $R²_w$(Δ)) and a technical duplicate baseline that together recalibrate performance assessment. Overall, the authors study the area of perturbation response modeling evaluation and propose using WMSE as a training objective, demonstrating reduced mode collapse and improved performance across multiple model architectures.

**Compliance With Llm Reviewing Policy:**

Affirmed.

**Final Justification:**

The rebuttal has sufficiently addressed the concerns brought up. I had some reservations regarding the technical soundness of the proposed WMSE objective and the associated DEG testing. The authors clarified strict separation between train and test for any of the calculations involved with this reformulated assessment, reducing the likelihood of any inadvertent data leakage. They also clarified the nature of using DEG for the WMSE, which is using the continuous t-statistics that accompany DEG analysis. The authors' argument that additional bootstrapping for the technical duplicate would make it computationally prohibitive is understandable, especially considering the current absence of ceiling baselines in the perturbation modelling field. The presence of such a baseline outweighs the statistical rigor of additional bootstrapping. The explanation of the foundation model scGPT provided in the rebuttal is helpful and should also be integrated into the paper. While Replogle2022 simulation results would strengthen the simulation results, I am convinced by the merit of the proposed method based on the empirical results on Norman2019 and Replogle2022 together with the simulated results derived from Norman2019. Because my concerns have been adequately addressed I raise my score from 4 (weak accept) to 5 (accept).

**Key Questions For Authors:**

**1. Weight computation during training:** How exactly are WMSE weights computed during training? When computing DEG t-scores "with respect to the rest of the perturbed cells in the dataset" (Section 3.3.3), does "the dataset" refer only to training perturbations, or does it include held-out test perturbations? Please confirm that no information from test perturbations (including their expression profiles or DEG statistics) is used when computing training weights. A clear description of the data flow would resolve this ambiguity.

**2. Technical duplicate baseline methodology:** The description of the technical duplicate baseline (Section 3.4) is quite minimal. For example, could you clarify if the random split performed once per perturbation, or across random subset of cells per perturbation (e.g., bootstrap)? Also, for perturbations with few cells (e.g., 64 in Replogle22, yielding only 32 cells per half), how does sampling variance affect the reliability of this baseline as a performance ceiling?

**3. DEG threshold effects:** The number of DEGs per perturbation varies dramatically (median 4 for Replogle22, 110 for Norman19). How sensitive are WMSE results to the statistical threshold used for DEG calling?

**4. Foundation model fine-tuning:** Section 3.5 mentions that scGPT was "fine-tuned for unseen perturbation prediction," but key details are missing. Was the pre-trained backbone frozen during WMSE training and a perturbation prediction head was tuned, or did gradients backpropagate through the entire model? Additionally, scGPT's original objective involves multiple loss components, how was WMSE integrated, and were other losses retained?

**Limitations:**

The paper does not include a dedicated limitations section, which is a missed opportunity given several important caveats to the work. Some examples are, firstly, the simulation study uses parameters derived from Norman19, but it is unclear how sensitive the conclusions are to this choice, would Replogle22-derived parameters yield similar results? Additionally, the technical duplicate baseline's reliability under low cell counts is not characterized, despite this being central to interpreting the proposed performance ceiling. Finally, the paper does not discuss potential failure modes of WMSE training, for instance, whether noisy DEG calling or perturbations with diffuse effects across many genes could introduce harmful biases.

**Strengths And Weaknesses:**

### Soundness

The paper is timely and addresses an interesting open question in the field: mean baselines outperform deep learning models for perturbation effect prediction with scRNA-seq data. The authors provide a principled perspective on this failure mode through the *signal dilution hypothesis*. They present a simulation study (10,000 synthetic datasets) that enables controlled analysis of parameter effects.

However, there is a critical methodological concern regarding information leakage in the WMSE training framework. The WMSE weights are computed using t-scores of DEGs vs Rest (Section 3.3.3), which requires knowledge of the perturbation-specific expression profile. In the unseen perturbation generalization setting, which is presented as the primary use case these models target, this information should not be available at time for held-out perturbations. At minimum, the authors should clarify whether any test perturbation statistics were included in the weight computation. Furhtermore, the technical duplicate baseline is a clever positive control that provides meaningful performance calibration. The proposed $R²_w$(Δ) metric is well-motivated, as computing deltas against $μ^{\text{all}}$ rather than $μ^c$ elegantly sidesteps control bias issues. The technical duplicate baseline (Section 3.4) is underspecified. Key details are missing: is the random split performed once or averaged over multiple partitions? How does sampling variance affect reliability when splitting yields only 32 cells per half (Replogle22)? Given that this baseline defines the proposed "performance ceiling," clearer characterization of its statistical properties is needed.

### Presentation
The paper is clearly written with an intuitive framing that effectively communicates the signal dilution problem. Figure 1 provides a helpful overview of the contributions. The mathematical notation is consistent and the experimental setup is well-described.

There is one minor factual error: CellOracle is characterized as a model using optimal transport, but it does not. It is actually based on gene regulatory network inference without any OT objectives.

### Significance
This work addresses an important open problem in perturbation effect prediction. The finding that current evaluation metrics systematically reward degenerate predictions has implications for how the field benchmarks perturbation response models. The suggestions on how to mitigate these issues with WMSE and $R²_w$(Δ) are significant contributions to advance the field.

The main limitation to significance is the unresolved question of how WMSE would work in realistic deployment scenarios. If WMSE requires perturbation-specific DEG weights that can only be computed with ground-truth data, its utility for training models intended for zero-shot prediction is unclear.

### Originality
The signal dilution diagnosis is a novel and useful contribution to understanding evaluation artifacts in this domain. The weighted metrics (WMSE, $R²_w(Δ)$) and the technical duplicate baseline are original proposals. The connection between metric artifacts and mode collapse during training is insightful, and while speculated on before, it has never been investigated in this level of detail.

---

> ### Author Rebuttal · Authors · 2026-03-31
>
> We thank the reviewer for their thorough assessment and are encouraged by their recognition of the impact of this work. We address each critique below:
>
> **Weight computation during training:**  As we mention in our response to reviewer DwD9, the weight computation for WMSE training weights excludes any test set perturbations, which are fully held out. This will be clarified in the final version of the manuscript.
>
> **Technical duplicate baseline methodology:** The technical duplicate is a single random split per perturbation, and no bootstrapping or repeated sampling is involved. It answers a specific question: "what would we observe if we sequenced additional cells from the same perturbation condition?" This design establishes a realistic positive control that incorporates experimental uncertainty, rather than serving as an estimate of experimental variance. We agree with the reviewer that evaluating performance across multiple splits would be valuable; however, this would require additionally bootstrapping both the DEG and NIR computations, making benchmarking computationally prohibitive, especially in the larger Replogle22 dataset.
>
> As a sanity check: even in the low-cell regime (Replogle22, Table S2), the variance of technical duplicate performance across perturbations is low (SEM two orders of magnitude below metric values). Since noisy within-perturbation performance would propagate into cross-perturbation variance, this suggests that the single-split estimate is stable and that stochasticity in the halving process does not materially affect the reliability of this baseline.
>
> **DEG threshold effect on WMSE:** WMSE weights are derived from the continuous t-statistic values of the DEG analysis, not from any binarized classification. The weighting is therefore independent of arbitrary significance thresholds.
>
> **Foundation model fine-tuning:** We followed the originally published scGPT setup for perturbation modeling (https://doi.org/10.1038/s41592-024-02201-0). Specifically, we fine-tuned the entire model (backbone included) for 10 epochs with a learning rate of 1e-4, batch size of 32, and a masked MSE loss. When training with WMSE, the only modification was replacing the uniform weights in the MSE objective with DEG-derived weights, and no other components or hyperparameters were altered. Full implementation details and code will be documented in our public repository upon acceptance.
>
> **Norman19 derived simulation parameters:** We acknowledge that deriving simulation parameters from a single dataset is a limitation. However, the consistency between our simulation findings and results on real data (including Replogle22, which was not used to calibrate the simulation) provides evidence that our conclusions generalize.
>
> **Minor notes:**
> 1. We acknowledge the CellOracle mischaracterization and will correct the introduction accordingly.
> 2. For realistic deployment scenarios, WMSE can be used to reduce mode collapse whenever there are sparse effects of covariates (cell type, perturbation, tissue, etc) in training data. Which means that it can be used both in (e.g. CellXGene) and perturbational (e.g. X-Atlas/Orion) pretraining settings.
> 3. The only place where we use p-value thresholds for DEG calling is Pearson($\Delta$) DEGs computation. In this case we follow the field's standards of $p<0.05$.
> 4. We appreciate the suggestion to discuss potential failure modes of WMSE. We agree with those the reviewer has highlighted and will include a dedicated discussion of these points in the final manuscript.

---

> > ### Author Rebuttal · Reviewer_r9z5 · 2026-04-02
> >
> > I would like to thank the authors for providing these clarifications and taking on board some of the proposed suggestions. The authors have sufficiently addressed the concerns I have raised. I raised my score from weak accept to accept.

---

> > > ### Author Response · Authors · 2026-04-03
> > >
> > > We thank the reviewer for their consideration and recognition of this study's value, and for raising their score in light of our clarifications.

---

### Official Review · Reviewer_DwD9 · 2026-03-12

**Soundness:** 3
**Presentation:** 4
**Significance:** 3
**Originality:** 3
**Overall Recommendation:** 4
**Confidence:** 4

**Summary:**

This paper examines the systemic biases in the evaluation and modeling of scRNA-seq perturbation response models and proposes corresponding improvements. The central observation is that perturbation-induced transcriptomic changes are highly sparse, with differentially expressed genes (DEGs) accounting for only 0.05% to 1.34% of the total transcriptome depending on the dataset. Because standard metrics such as MSE and Pearson(Δ) aggregate error uniformly across all genes, the vast majority of unaffected background genes dominate the calculation, making it trivially easy for a degenerate predictor—one that simply outputs the dataset-wide mean expression profile—to achieve competitive scores without learning any perturbation-specific information. The authors further show that Pearson(Δ), being computed relative to a control population, is additionally susceptible to control-cell bias, which can independently inflate the apparent performance of this mean baseline.

To address these issues, the authors reconstruct both the evaluation and training framework around DEG-aware weighting and multi-level baseline calibration. On the metric side, they introduce WMSE, which up-weights perturbation-specific DEGs identified via t-tests against all other perturbations, and a weighted delta R² (R²_w(Δ)) that measures changes relative to the mean of all perturbed cells rather than the control group. The latter has the desirable mathematical property that any constant mean prediction yields a value strictly ≤ 0, cleanly placing the mean baseline at null performance. For calibration, the authors propose three reference points: the control mean as a biased negative baseline, the dataset mean as a null baseline, and a novel technical duplicate baseline constructed by splitting cells from the same perturbation in half and using one half to predict the other, which defines a realistic performance ceiling given the inherent experimental noise. Finally, WMSE is proposed as a drop-in replacement for the standard MSE training objective, and experiments across GEARS, scGPT, scLambda, and FMLP show that it substantially reduces mode collapse—in GEARS on Replogle22, the fraction of ground-truth variance recovered increases from 14.51% to 40.56%—with the most pronounced gains observed for stronger perturbations.

**Compliance With Llm Reviewing Policy:**

Affirmed.

**Final Justification:**

I keep my score as "weak accept". The rebuttal addressed most of my questions but not fully answered the concerns related to WMSE on weak perturbations with clear numbers.

**Key Questions For Authors:**

1. The WMSE training objective requires perturbation-specific DEG weights, which are derived from ground-truth expression profiles. For unseen perturbation tasks, these ground-truth profiles are by definition unavailable at training time. How exactly are the weights computed during training—are they derived solely from training-set perturbations, or is any information from held-out perturbations used? If the latter, this would constitute information leakage that could inflate the reported gains. Please clarify the precise weight computation procedure for both the unseen gene and unseen combo settings.
2. The proposed metrics and training objective rely heavily on DEG identification via t-tests. Since DEG calls are sensitive to the choice of statistical test and significance threshold, have you assessed whether model rankings remain stable when alternative methods (e.g., Wilcoxon test or DESeq2) or different p-value cutoffs are used? A sensitivity analysis here would strengthen confidence in the framework's robustness.
3. For perturbations with zero significant DEGs, how does the weight computation procedure behave in practice—does it fall back to uniform weights, and if so, is WMSE effectively identical to MSE for these samples? More broadly, is there a risk that WMSE training causes the model to systematically deprioritize weak perturbations in favor of strong ones, potentially degrading performance on the former?

**Limitations:**

The authors not adequately discussed the limitations of their proposed methodology. I recommend adding the following point in the limitation sections.
Sensitivity: The robustness of WMSE and weighted R-squared to the specific DEG calling algorithms and statistical thresholds used.
Zero-Shot Applicability: The practical challenge of estimating DEG weights for completely unseen perturbations where ground-truth expression data is unavailable.
Weak Signals: The potential risk of the WMSE training objective ignoring or suppressing weak perturbations that lack statistically significant DEGs (a phenomenon hinted at in Supplementary Fig. S5 ).

**Strengths And Weaknesses:**

Strengths:
1. Accurate problem identification with rigorous attribution analysis. The paper accurately identifies signal dilution and control bias as the root causes of inflated mean baseline performance. Crucially, the authors go beyond hypothesis by conducting large-scale in silico simulations alongside real-world experiments, systematically varying dataset parameters to quantify each factor's contribution. This controlled attribution analysis substantially strengthens the validity of the claims.
2. Biologically grounded evaluation framework. The use of DEGs vs. Rest (rather than DEGs vs. Control) for weight calculation fundamentally severs the control bias pathway that confounds traditional metrics. Computing R²_w(Δ) relative to the perturbed mean rather than the control further eliminates reference-dependent inflation. Together, these design choices ensure the metrics capture genuinely perturbation-specific signals. The technical duplicate baseline is also a valuable addition, providing the field with a previously absent meaningful performance ceiling.

Weakness:
1. Limited effectiveness on weak perturbations. In the weakest DEG quantiles (0–25%), WMSE training shows unstable or even negative gains (Table S3). Since perturbations with few or no DEGs cause the weights to degenerate toward uniformity, WMSE and MSE become nearly indistinguishable in exactly the most challenging regime—which constitutes the majority of perturbations in datasets like Replogle22. This limitation deserves more direct discussion.
2. Potential data leakage in WMSE weight computation. The per-perturbation DEG weights rely on t-scores computed against all other perturbations, which requires access to ground-truth expression profiles. While this is unproblematic at evaluation time, the paper does not clarify how weights are obtained during training—specifically, whether test perturbation expression data is used. If so, this constitutes information leakage that could inflate the reported training benefits.

---

> ### Author Rebuttal · Authors · 2026-03-31
>
> We thank the reviewer for their recognition of the impact of this work, and also for the opportunity to strengthen our work by clarifying key methodological details and addressing important limitations.
>
> **WMSE weight computation protocol:** When computing weights for WMSE, we take a separate approach for different splits. For train and val, the “rest” reference for DEG computation was restricted to train and val perturbations, preventing information leakage from the held-out test set perturbations into training loss weights. For the test set, all the dataset (train/val/test) is taken as reference for DEG computation, which is unproblematic as noted by the reviewer and increases metric quality. We fully agree with the reviewer's suggestion to provide clarity on this point and will include an updated data splitting description in the final version.
>
> **DEG methods:** Regarding the effect of DEG computation methods on metric/model outcomes, we performed an ablation study in the Norman19 dataset with all methods available on Scanpy. Results in the NIR metric (calibrated and not depending on weights) show the t-test with overestimation of variance as a marginal second to the Wilcoxon test. Taking into account the computational efficiency of the t-test compared to Wilcoxon (which can become intractable on larger perturb-seq datasets) we find this is an acceptable trade-off between performance and computational complexity. We are happy to include this experiment with appropriate discussion in the camera ready version of the manuscript.
>
> On a related note, we want to clarify that our results do not depend on any DEG p-value cutoffs since all weights are defined with a continuous functional form based on the test statistic from the t-test (Sec. 3.3.3).
>
> **Is WMSE systematically deprioritizing weak perturbations?:** Regarding the possibility of deprioritizing weak perturbations with the WMSE loss, any loss defined for this task (including MSE) is likely to prioritize stronger perturbations over weak ones since they drive bigger supervision errors. Consequently, we might consider a complementary question to that posed by the reviewer: “does WMSE deprioritize weak perturbations more than MSE?”. We think the answer to this question can not be drawn from the test results in our manuscript but would require a stratified analysis of training losses. Stratifying the training loss by perturbation strength might reveal whether reducing error on strong perturbations comes at the cost of increased error on weak ones. We plan to investigate this and, if feasible, include the results in the appendix of the final manuscript.
>
> **Minor notes:** WMSE does not fallback to regular MSE if there are no DEGs. When a perturbation has no DEGs, we follow the same protocol defined in Sec 3.3.3 meaning that the weights are defined continuously from the t-statistic of DEG analysis. Thus they are still prioritized according to their deviations from the whole population even in the case that no DEGs meet the significance cutoff.

---

> > ### Author Rebuttal · Reviewer_DwD9 · 2026-04-03
> >
> > The authors have addressed most of my concerns except for the weak perturbations question. The author promised including additional ablation studies in their final version instead of presenting one or two sample results in their rebuttal, thus I keep my original score.

---

### Official Review · Reviewer_eoPY · 2026-03-18

**Soundness:** 3
**Presentation:** 3
**Significance:** 2
**Originality:** 2
**Overall Recommendation:** 4
**Confidence:** 3

**Summary:**

The paper is focused on predicting single cell transcriptomic effects of gene perturbations. This predictive problem has recently seen introduction of advanced machine learning models, yet there is no consensus on how to measure the effectiveness of predictive models given stability of expression of many genes, and perturbation effect often resulting in changes with narrow support. The paper proposes a method that introduces gene weights int $R^2$ and $MSE$ based metrics.

**Compliance With Llm Reviewing Policy:**

Affirmed.

**Final Justification:**

The paper focuses on an existing problem in gene perturbation effect prediction. The paper is well written, and while the focus is narrow in scope, it furthers our understanding of the strong performance of simple baselines. Additional information about WMSE training provided in the rebuttal has increased the significance of the paper.

**Key Questions For Authors:**

1. Does training with WMSE impact generalization to unseen cell lines?
2. Does it also improve more recent methods such as TxPert (Wenkel et al., 2025)?

**Limitations:**

yes

**Strengths And Weaknesses:**

The paper is written in an accessible way, but is mostly on explaining the measurement problem and exploring the properties of the metric. That contribution is relatively narrow, and the proposed metric that incorporate weights based on differential expression are not particularly novel, since limiting results to a smaller set of DEG is a common practice in the field. The contribution that has the highest potential to impact the field, using the new metrics to improve model training, is a relatively underdeveloped part of the manuscript in its present form. The manuscript would have higher significance if the effect of WMSE training was investigated more in depth, analyzing aspects such as generalization not only to unseen perturbations, but also to unseen cell lines.

---

> ### Author Rebuttal · Authors · 2026-03-31
>
> We would like to thank the reviewer for their insightful comments which we have addressed below:
>
> **Novelty:**
>
> Our focus on the measurement problem is intended to address an ongoing debate in the field.  Recent studies (e.g., https://doi.org/10.1038/s41592-025-02772-6) have argued that degenerate predictors (e.g., the “mean baseline”) outperform trained deep learning models, raising questions about the utility of perturbation response modeling overall. Our work sheds new light on this debate and introduces additional contributions to perturbation response modeling with (1) a simulation study examining the factors driving mean baseline performance, highlighting signal dilution as a key metric artifact, (2) DEG-aware metrics that address issues of signal dilution through an error weighting strategy, and (3) a proof-of-concept demonstrating that one of these new metrics (WMSE), when used as a training objective, mitigates mode collapse and improves model performance (including in orthogonal, unweighted metrics). We provide further discussion of each below and argue for their value as novel contributions:
>
> *Simulation study:* By isolating dataset factors driving mean baseline performance, we identify metric artifacts (most notably signal dilution) that can confound the evaluation of perturbation response models. As acknowledged by other reviewers, several of these effects had been hypothesized but never rigorously tested, while others constitute new observations about the statistical properties of Perturb-seq data that will inform future modeling efforts.
>
> *DEG-aware metrics:* The reviewer is correct that DEG-filtered evaluation is established practice. Our contribution is orthogonal to filtering: First, rather than using a control cell reference for DEG calculation, we instead used the rest of the perturbed cells in the dataset. This modification accentuates the signals that make each perturbation unique rather than the signals that merely differentiate perturbed cells from unperturbed controls (which can be the source of another metric artifact: “control bias”). Second, we derive continuous weights from DEG test statistics rather than applying a binary top-k filter. This allows all genes to contribute to the evaluation of model performance, mirroring the goal of modeling perturbation effects across the transcriptome and not just in the top DEGs.
>
> *WMSE as a training objective:* Our proof-of-concept study demonstrates that WMSE reduces mode collapse across four architecturally distinct models. We appreciate the reviewer's recognition of this as the highest-impact contribution and are committed to deepening this analysis in the final manuscript. Specifically, we will include the results of an ablation study on alternative weight definitions and DEG calling methods. Briefly, the results of this analysis support our design choices for the DEG computation method and the functional form used in this work to calculate the WMSE weights.
>
> Taken together, these contributions address an ongoing methodological debate through systematic examination of factors driving mean baseline performance, leading to new domain-specific insights, and introducing a training objective with benefits across multiple modeling paradigms.
>
> **Generalization to unseen cell types:** In this work, we focus on addressing the questions raised by the recent reports of strong mean baseline performance when compared to trained deep learning models. Because those reports focused on unseen single perturbations and unseen combinations, we examined the same task types in the present study. We fully acknowledge the importance of cross-cell-type generalization and note that, regardless of the prediction task, WMSE is a loss designed to counteract mode collapse and improve model performance. Thus, we expect it will confer benefits in any setting where models exhibit this failure mode, including unseen cell types.
>
> **Newer models:** We appreciate the value of broad model coverage and we feel TxPert is important to discuss as it is a recent preprint relevant for our background discussion regarding benchmarking baselines (this will be included in the final version of the manuscript). However, our primary aim in this work is to diagnose a specific metric and modeling artifact raised by recent benchmarking studies, rather than evaluating models more broadly ourselves. To validate the generality of our proposed solution, we selected models spanning four distinct architectural paradigms: GEARS (graph-based), scGPT (foundation model), scLambda (VAE), and FMLP (decoder-only with gene embeddings). This diversity of model families provides multiple lines of evidence that the observed improvements are not architecture-specific, even without more extensive analysis of additional models.

---

> > ### Author Rebuttal · Reviewer_eoPY · 2026-04-03
> >
> > Thank you for the additional information and additional results on WMSE-based training. I have raised my score from 3 to 4.

---

### Decision · Program_Chairs · 2026-04-30

**Decision:**

Accept (regular)

**Comment:**

Reviewers unanimously recommend accept (most weak).  The signal-dilution diagnosis and DEG-aware metrics are a timely, well-executed contribution, though evaluation is limited to two real datasets and the WMSE training study is proof-of-concept.